# Product preferences and willingness to pay for potable water delivery: Experimental evidence from rural Bihar, India

**Drew B. Cameron**[1,2]*, **Isha Ray**[3], **Manoj Parida**[4], **William H. Dow**[2]

1 Department of Health Policy and Management, Yale School of Public Health, New Haven, Connecticut, United States of America, 2 Division of Health Policy and Management, Berkeley School of Public Health, University of California, Berkeley, Berkeley, California, United States of America, 3 Energy and Resources Group, University of California, Berkeley, Berkeley, California, United States of America, 4 DCOR Consulting, Bhubaneswar, Odisha, India

* drew.cameron@yale.edu

**Data Availability Statement:** All underlying data files for this manuscript will be available from the Harvard Dataverse (https://doi.org/10.7910/DVN/YZOSVS).

## Abstract

Despite dramatic reductions in global risk exposures to unsafe water sources, lack of access to clean water remains a persistent problem in many rural and last-mile communities. A great deal is known about demand for household water treatment systems; however, similar evidence for fully treated water products is limited. This study evaluates an NGO-based potable water delivery service in rural Bihar, India, meant to stand-in for more robust municipal treated water supply systems that have yet to reach the area. We use a random price auction and discrete choice experiment to examine willingness to pay (WTP) and stated product preferences, respectively, for this service among 162 households in the region. We seek to determine the impact of short-term price subsidies on demand for water delivery and the extent to which participation in the delivery program leads to changes in stated preferences for service characteristics. We find that mean WTP for the first week of service is roughly 51% of market price and represents only 1.7% of median household income, providing evidence of untapped demand for fully treated water. We also find mixed evidence on the effect of small price subsidies for various parts of the delivery service, and that one week of initial participation leads to significant changes in stated preferences for the taste of the treated water as well as the convenience of the delivery service. While more evidence is needed on the effect of subsidies, our findings suggest that marketing on taste and convenience could help increase uptake of clean water delivery services in rural and last-mile communities that have yet to receive piped water. However, we caution that these services should be seen as a stopgap, not a substitute for piped municipal water systems.

## 1. Introduction

The World Health Organization estimates that roughly 2 billion people globally use a drinking water source that is contaminated with feces, and that polluted drinking water sources contribute to nearly half a million untimely deaths per annum [1]. Access to piped water has expanded

**Funding:** DBC; IND-19059; the International Growth Centre; https://www.theigc.org; The funders had no role in study design, data collection and analysis, decision to publish, or preparation of the manuscript. DBC; Center for Global Public Health, University of California, Berkeley, School of Public Health; https://cgph.berkeley.edu; The funders had no role in study design, data collection and analysis, decision to publish, or preparation of the manuscript.

**Competing interests:** The authors have declared that no competing interests exist.

significantly [2], but, as of 2020, just over 83% of urban households worldwide and 42% of rural households received piped water services [3]. In the absence of government-managed or -regulated water services, small scale private providers and NGOs have attempted to provide low-cost fully treated water deliveries in low-income settings [4, 5]. These deliveries are not a substitute for robust utility-scale water systems [6] but have value as interim approaches to safely managed water. Delivered water systems bring safely managed water closer to a service model than bottled water purchases from stores or kiosks; they eliminate the hassle costs of transporting water, which are especially burdensome for families without reliable means of transportation.

The market for community-based and NGO-supplied clean water production is changing quickly in rural India. In a household water usage survey in rural Bihar, Brouns and colleagues [7] find that: 1) village residents believe that the government should be responsible for the provision of safe and free water; 2) the abundance of free shallow well water is likely responsible for relatively low willingness to pay for treated water products; and 3) the three most important factors in choosing a water source that is free of pollutants are that it is "clean, tasty and simple to use" [p12]. In a study of water use in West Bengal, Delaire and colleagues [4] find that the strongest determinants of purchased versus well-water use are socioeconomic status, perceived likelihood of GI illness and dissatisfaction with iron taste. Although many studies already examine willingness to pay (WTP) for water treatment products [8, 9], as well as the concomitant burden of water collection time [10, 11] and quality [12], very little evidence exists for fully treated water delivery in low-income rural communities. Studies have also found that providing subsidies for the up-front cost of in-home piped water are successful at encouraging uptake [13], but similar research on a regularly delivered packaged water is missing.

In this paper we evaluate user preferences and willingness to pay for treated water deliveries in the Supaul district, rural Bihar, India. As of 2020, 85.4% of rural households in the study region use ubiquitous (mostly government-built) shallow well hand pumps as their primary drinking water source [14]. This represents a slight improvement over the 93% who primarily used this source less than a decade earlier [15] as access to municipal water systems has improved. Indeed, nearly all (99%) rural households have access to these or better sources within their household plots, meaning that time spent collecting water is minimal, and most (94.2%) report using no water treatment [14, 15]. Despite their convenience, shallow wells are subject to inundation from polluted surface water during periods of heavy rainfall, introducing the risk of biological contaminants. Further contamination from agricultural runoff presents health risks to young children and the presence of high levels of heavy metals such as iron can lead to an unpleasant taste and corrosion in any existing water infrastructure as well as incrustation of water piping surfaces like those in existing pumps, providing a locus for biofilms that can harbor harmful microorganisms [16–18].

This study has multiple aims designed to inform future WASH studies and programs on demand and preferences for clean water. The first is to examine willingness to pay (WTP) for a treated water delivery service among residents to assess product pricing both in real terms and as a share of household income to inform the literature on product pricing. Second, we seek to examine the possible role of subsidies in increasing short-term demand and product uptake to inform a larger, randomized controlled trial that examines the potential for subsidies to lead to price anchoring and/or positive learning. Third, we aim to identify specific product characteristics that are subject to experiential learning among new customers to add to existing knowledge on how preferences for water products change over time. In regions where public utilities are unlikely (for the present) to enter "last-mile" communities, a more complete understanding of the acceptability and affordability of treated and delivered water is essential for policy-making on safe drinking water solutions.

## 1.1 Willingness to pay, subsidies and learning effects

A growing body of literature examines willingness to pay (WTP) for a variety of water treatment products [19, 20] in low-income rural settings. Others examine the concomitant impact of subsidies on WTP for and adoption of products, including household water filters and treatment solutions [21, 22] and fully treated water sachets [23]. These studies regularly find that lower prices *may* increase demand for and use of clean water products, but that such outcomes are dependent on *a priori* willingness to pay (a.k.a. 'screening effects'), subsidy size relative to baseline product use, general knowledge about the market price of the product, the burden of product maintenance, the influence of social pressures and the frequency of reminders about the importance of use.

Learning effects broadly refer to the process of gathering information about a particular product or process through experience after purchase. The direction of this learning is theoretically ambiguous (can be positive and/or negative with respect to product acceptance). It can take place through a combination of direct experience [19, 24–28] and social learning via peer interactions and observations [27, 29–31]. Several studies of clean water products investigate preferences for products and product characteristics [19, 20, 32–34]. Others examine processes of social learning about different types of treatment products [19]. Among these studies, some of the most important water product characteristics are price, taste, smell, health benefits (both real and perceived), convenience of use (including time required), durability and aesthetics. Preferences for these characteristics are sensitive to whatever alternative sources of water exist and to community and social pressures.

We hypothesized that random price auctions would help to uncover any latent local demand for water delivery and sought to determine the extent to which subsidies could be leveraged to increase uptake. After adoption, we further anticipated scope for positive learning about several characteristics of the water delivery service including the taste of the water (alternative ground water sources are heavily iron-polluted), convenience of delivery, and perceived health benefits. We also anticipated that the scope for positive learning around these characteristics could be amplified through a social marketing campaign provided by residents, social learning among neighbors, repeat reminders and product offers. Meanwhile, the scope for negative learning seemed possible for the convenience of daily deliveries (in the case of delayed or missed visits), the temperature of the water (when stored water warms on hot days or when delivery is not prompt) or real health outcomes (alternative routes of infection could render fidelity of clean water use insufficient to avoid exposure).

# 2. Materials and methods

## 2.1 Site selection and sampling

In the absence of municipal water services in parts of rural Bihar, several local providers offer sale of safe water. One organization, Sanitation Health Rights in India (SHRI), has been providing water and sanitation services in the Supaul district of Bihar since 2013 and maintains a water delivery service for several hundred local customers. SHRI water is generated using a patented *Drinkwell* water system that pumps deep groundwater through a carbon-filtration and UV treatment system followed seasonally by an air conditioning unit to produce clean, chilled potable mineral water. The water is tested quarterly to ensure the removal of iron, fluoride, and arsenic as well as all biological contaminants. SHRI then sells water to communities, delivered on 3-wheeled vehicles in 1000-liter tanks, which fill 20-liter bottles owned by customer households for a per-unit price of ₹10 (or $0.14; ₹71 = USD 1 in 2019). The per unit price sometimes rises as high as ₹15 ($0.21) during summer months when the plant uses a

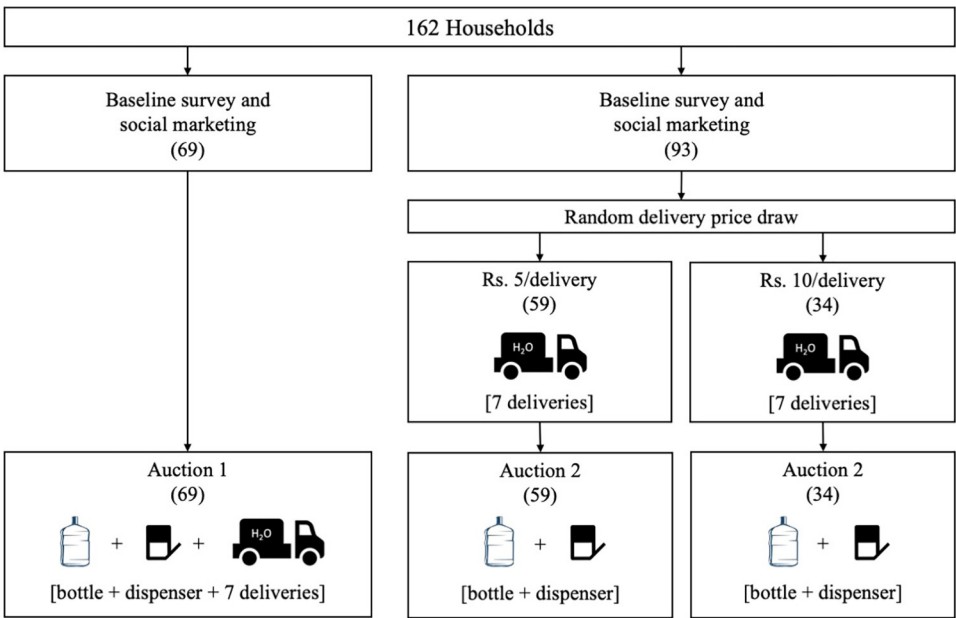

**Fig 1.** CONSORT diagram–survey households allocated into Auction 1 for combined purchase of bottle, dispenser and deliveries (left) or Auction 2 for purchase of bottle and dispenser only (right); number of households in parentheses.

chiller to cool the water prior to delivery. Reverse osmosis water produced and delivered by other vendors usually retails for between ₹15 and ₹20 in the same neighborhoods. Before 2019, SHRI customers paid an additional (refundable) ₹120 deposit for each bottle and dispenser (see **Fig 1**). However, because of high rates of theft, damage and loss of these bottles and dispensers, the hardware was sold to participating households starting in early 2019. By the time of this study, all SHRI customers of the water service had to purchase a bottle and dispenser (the hardware) either from SHRI at a wholesale price of ₹250, or from another vendor in the market where a single bottle and dispenser normally retails for ₹275. These bottles and dispensers are ubiquitous in the area and used as the primary water storage device for those purchasing treated water from any local vendor. Their normal retail price is widely known. Notably, we also found that these storage bottles were sometimes repurposed by customers who discontinued service to store grains and other dry foods to prevent spoiling from mold.

**Fig 1** is a CONSORT diagram of the study stages described in sections 2.1 through 2.3. Working closely with SHRI, we identified participant households within a geographic area that was easily reachable by program staff, enumerators, and delivery drivers (i.e., on the road system and adjacent to the NGO's newest water plant). We identified 162 households in five small neighborhoods that were roughly representative of the potential customer base of the NGO in terms of income level and occupation. Participants could not be current SHRI water users, had to be allowed to make purchase decisions for the households, and had to be willing to participate in study activities. Households were defined as a family unit using a common kitchen. Our sample of 162 households were nested within 112 physical structures (thus, multiple households could reside within the same structure). Participants gave voluntary and informed consent to the study activities including baseline and follow-up survey interviews, a social marketing exercise and the price auction "game" in which respondents would bid on the price of water delivery. Our research protocol was approved for ethical compliance by the University of California Berkeley's Office for the Protection of Human Subjects (protocol #2018-

04-11016). Verbal consent was obtained from study participants due to high rates of illiteracy among the study population. The study team was also working in partnership with the registered Indian NGO SHRI. As a registered Indian NGO, SHRI has blanket approval from the Government of India to conduct surveys among its customers. The original intent of this study was as a pilot to conduct a random price auction for the price of water only and examine demand for water after subsidies expired (that study design is preregistered with AEA's RCT Registry [35]). However, rapid cycle changes on the ground led to design modifications that disallowed the intended study to be completed. Instead, we undertook an auction for the combined package of services during the first two days of data collection and then made another mid-course change to randomize the price of water (₹10 vs. ₹5/delivery) before respondents bid on only the hardware.

**2.1.1. Inclusivity in global research.**   Additional information regarding the ethical, cultural, and scientific considerations specific to inclusivity in global research is included in the Supporting Information.

**2.1.2. Study setting.**   The setting for this study was characterized by small single- and multi-family dwellings not connected to municipal water and located adjacent to paved and gravel roads. Reported median monthly income was ₹9000 ($121.62). Most households, 77.8%, were Hindu while 22.2% were Muslim. Despite being connected to the road system, compared to the general population of rural Bihar, our sample was slightly less socioeconomically advantaged. **Table 1** compares several key demographic and household characteristics between households in this study and those in wider rural Bihar. Most notably, several indicators are worse than average for rural Bihar, including the level of education of household heads (compared to the same statistic for *all* men and women in Bihar), the percent of households practicing open defecation, percent of household structures made of improved materials (floors, walls and roofs), and the percent of children under 5 having experienced diarrhea in the past two weeks [36]. Because this sample is slightly poorer than the average Bihari household, our results may have a slight downward bias with regard to product adoption, though they would likewise not be generalizable to the poorest of the poor households in the region.

Despite being poorer on average, water use patterns among study participants are well in-line with normal practices for rural Bihar. All study households reported primarily using hand pumps with shallow wells as their principal source of water for drinking, cooking, and bathing. As with the rest of Bihar, the status quo time burden of water collection in our sample is minimal with all but 2 households having primary water sources in their homes or yards. In total, 17.9% of households self-reported ever treating their water before use–almost all of whom used boiling as the primary treatment strategy [14, 36]. Since water is collected a la carte by almost all households, we do not have data on the daily total volume of water collected or consumed by households.

## 2.2 Social marketing

To encourage uptake of the water delivery service, SHRI's executive director identified five young men between 18 and 25 years of age to be trained as social marketers and to join the survey enumerators after each baseline survey was completed. For each of the 162 households surveyed, these social marketers gave a safe water demonstration designed to impart information about the safety and quality of SHRI water, share their experiences with clean water use, and provide a free taste test of the treated water. Inspired by previous research showing that tests of fecal contamination led to increased purchase of treated water [37], the marketing team gave a visual demonstration of the iron content of the water. In each home, the marketing team filled one clear plastic cup with treated water and another with water from the family pump. Guava

**Table 1. Sample characteristics of study households versus the rest of rural Bihar.**

| | Study sample | Rural Bihar |
|---|---|---|
| Religion of household head: Hindu | 77.8 | 86.5 |
| Religion of household head: Muslim | 22.2 | 13.3 |
| Female head has any formal education | 15.5 | |
| Male head has any formal education | 51.9 | |
| % of females with any education | | 58.8 |
| % of males with any education | | 77.7 |
| Household keeps farm animals | 84.6 | 63.4 |
| Household practices open defecation | 74.1 | 43.9 |
| Household has electricity | 98.0 | 95.6 |
| Improved household structure (roof, walls, floor; "*pucca*")[$$] | 17.9 | 26.9 |
| Children under 5 who experienced diarrhea in the past two weeks | | |
| % of all study households | 26.5 | |
| % of only study households with children <5 | 45.0 | |
| % all children <5 (Bihar) | | 13.9 |
| % all children <5 (Supaul)[$] | | 39.3 |
| Water source on premises | 98.8 | 88.4 |
| Source: piped water to home/yard/plot | 0.0 | 7.5 |
| Source: piped water to neighbor | 0.0 | 1.2 |
| Source: piped water to public standpipe | 0.0 | 2.5 |
| Source: tube well borehole | 100.0 | 85.4 |
| Report treating water before use[$$$] | 17.5 | 5.8 |
| water treatment strategy: boiling[$$$] | 17.2 | 2.1 |
| water treatment strategy: stand and settle[$$$] | 0.6 | 0.6 |

Notes: Questions may differ slightly between this sample survey and Bihar DHS data [14]; $ [36]; $ $ *pucca* indicates improved roof, walls, and floor materials in DHS survey; $ $ $ Study survey asks respondents if they "ever treat water before drinking," DHS states respondents "treat water prior to drinking"

leaves were then crushed by hand and placed into each cup. While the color of the treated water remained mostly unchanged, iron in the untreated water reacted with tannins in the leaves to turn the water dark within roughly 2 minutes (see: **Fig 2**). This procedure was meant to demonstrate the relative purity of the treated water for sale. Respondents could replicate the procedure if they wished, after which time the social marketing team conducted a question-and-answer session for each household before departing the survey site.

## 2.3 Willingness to pay: Random price auctions

Once the social marketers had left the interview site, study enumerators conducted random price auctions (framed as "games") with all 162 households. Based on Becker et al. [38] and modified from Burt et al. [20] and Berry et al. [22], these auctions were intended to elicit willingness to pay for a week-long delivery scheme that included the purchase of a bottle and dispenser. The purchase of hardware was required to become a customer. There were two auction types which each followed a specified script (see: Appendix 1 in S1 File). The first auction was conducted with the first 69 households that were identified during the household selection process. Respondents were asked to bid on a combined package of hardware and seven water deliveries (retailing from SHRI at ₹10 per 20-liter delivery). Thus, the initial auction was for a combination of these products. A second auction was devised mid-study and conducted with the remaining 93 households identified later during the listing procedure.

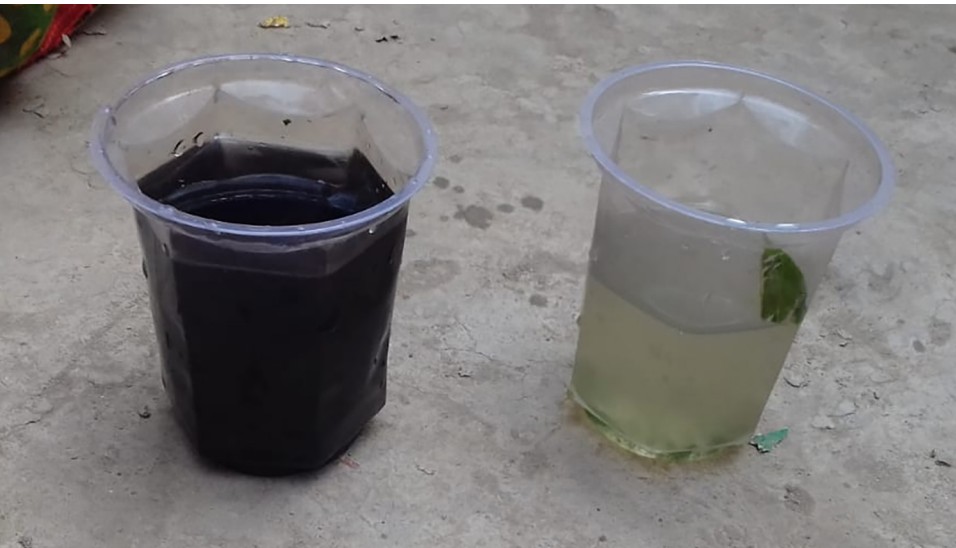

**Fig 2.** Treated water demonstration–Drinking water 2-minutes after crushed guava leaves are added to untreated ground water (left) versus treated ground water (right).

Participants first drew a number from a bag (either ₹5 or ₹10), which signified the per-delivery price they would face for seven water deliveries (1 week; totaling ₹35 or ₹70, respectively) should they win the auction. The "per-delivery" price was for filling one bottle with treated water. In total, 59 out of the 93 households in Auction 2 (63.4%) drew the reduced per delivery price of ₹5 per delivery while the remaining 34 households (36.6%) drew ₹10. Knowing the per-delivery price they would face if they won the subsequent auction and chose to purchase, respondents in Auction 2 were then asked to bid on the value of the hardware only (bottle and dispenser). Winners of Auction 2 could then choose to purchase the hardware and 7 water deliveries at the per-delivery price initially drawn. In either auction, households *could have* purchased more than one delivery per day or could elect to have seven deliveries spread over more than seven days. In either case a maximum of seven filled bottles were allowed among winners. No households elected to receive more than one delivery per day.

In both auctions, participants were first asked to state a price they were willing to pay for the product (the bottle, dispenser and seven deliveries in Auction 1; the bottle and dispenser only in Auction 2). Respondents then selected one of thirteen envelopes from a paper bag, blinding them to the available prices. Each envelope contained a piece of paper with a different price value that was less than or equal to the retail price of the combination of products on offer in either auction, not including a zero-price. If the price drawn from the bag was less than or equal to the price they had stated, the respondent "won" the auction and could purchase the product for the price drawn. If the price drawn from the bag was greater than their stated price, the respondent "lost" the auction and could not purchase the product. This scenario gave respondents an incentive to state the highest price they would be willing to pay if they were interested in maximizing their chances to "win" the auction. The rules of the game were explained to all participants in advance, and each participant was given at least one practice round to bid on a product of lesser value (a bar of soap, following Burt et al. [20]) so they could run through the mechanisms of the game (auction procedures are reproduced in Appendix 1 in S1 File).

Households were selected for the study during a listing process in which members of the study enumeration team walked through eligible neighborhoods and screened potential

households for inclusion. Eligible households were not selected into Auction 1 versus Auction 2 by randomization, but rather by listing order, thus it is possible that household characteristics differ between the two groups, which could potentially introduce bias of uncertain sign in our comparisons of the auctions. Table 3 presents formal tests of differences in baseline characteristics between the two auction groups, and finds significant differences, indicating the importance of testing for sensitivity of results to controlling for these baseline characteristics so as to quantify potential bias at least from these observed differences. In Table 5 we present the main auction comparisons without versus with these baseline controls and find that adding these controls has only minor effects on the estimated coefficients and does not change our overall conclusions.

**2.3.1 WTP research questions.** These auctions provide estimates of willingness to pay for i) the combined delivery package (hardware + seven water deliveries) ii) the requisite hardware, given a 50% discount on the price of seven water deliveries, and iii) the requisite hardware, given no discount on the price of seven water deliveries. To determine WTP for hardware and the first week (or 7 deliveries) of the water service, we examine average bid prices among respondents in Auction 1. To determine willingness to pay for the hardware alone as a prerequisite for participation in the delivery service, we separately examine mean bid prices in Auction 2.

In socio-demographically balanced groups, the comparison of the two auctions presents a unique opportunity to examine the effect of introducing random promotional discounts on willingness to pay (in Auction 2) relative to no promotional discounts being offered (in Auction 1). This could help to identify the effects of a promotional discount on WTP among those receiving the discount. Specifically, we ask: *Compared to WTP for the whole delivery service (Auction 1), does the result of receiving a promotional discount for water deliveries (in Auction 2) impact willingness to pay for the requisite hardware*? To answer this, we test the following hypotheses as represented in **Fig 3**. In this figure, x represents the willingness to pay for the whole delivery service as a package (bottle + dispenser + deliveries of water). The value *a* represents the change in demand placed on the overall package that results when consumers are met with an unexpected discount of ₹35 on the price of water deliveries–or the effect on demand of *winning*. Meanwhile, the value *b* represents the corresponding change in demand for the overall package that results when consumers knowingly *do not win* a discount of ₹35 on the price of water deliveries–or, the effect on demand of *missing out*.

$H_0$ –If promotional discounts on water deliveries do not affect willingness to pay for requisite hardware, we expect to see average bid prices on hardware in Auction 2 equivalent to average bids in Auction 1 ($x$), less the price that will be paid for seven water deliveries (i.e., $a = 35$ in the discount group; and $b = 70$ in the no discount group).

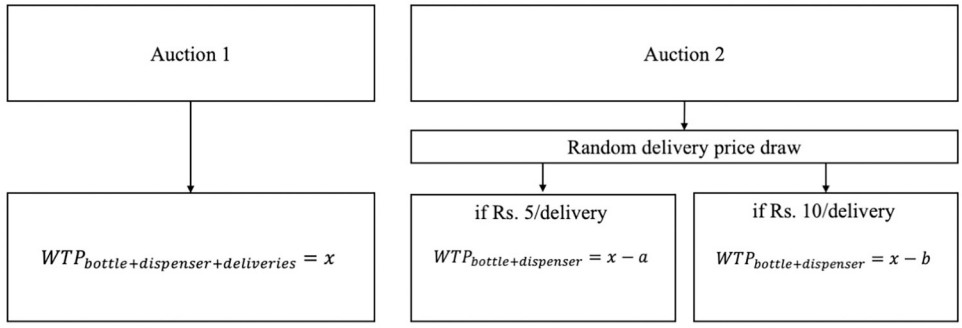

**Fig 3. Decision tree for willingness to pay (WTP) under each auction.**

$H_1$ –If promotional discounts on water deliveries increase WTP for the requisite hardware, we expect to see a corresponding difference in bid price for hardware that is less than ₹35 in Auction 2 compared to Auction 1 (i.e., $a < 35$). We would expect this if the effect of winning has a disproportionately large positive impact on demand for the delivery package.

$H_2$ –If "missing out" on promotional discounts on water deliveries decreases WTP for the requisite hardware, we expect to see a corresponding difference in hardware bid price greater than ₹70 in Auction 2 compared to Auction 1 (i.e., $b > 70$). We would expect this if the effect of missing out has a disproportionately large negative impact on demand for the delivery package.

Thus, $H_1$ represents the size of any possible price effect on WTP for hardware, while $H_2$ represents the size of a possible negative externality on WTP for hardware. Our primary analysis uses ordinary least squares regression, where for household $h$, within structure $s$, in neighborhood $n$, $y_{hn}$ is the average bid price of hardware (bottle + dispenser), $\beta_0$ is the intercept bid price in the combined auction group, $water\_price_{hn}$ is the random draw of either discount or no discount on the price of water, $X_{hn}$ is a vector of household-level covariates, $\eta_n$ is a vector of neighborhood-fixed effects, and $\epsilon_{hs}$ is an error term clustered at the structure-level.

$$y_{hn} = \beta_0 + \beta_1 water\_price_{hn} + \beta_2 X_{hn} + \eta_n + \epsilon_{hs} \tag{1}$$

**2.4 User preferences and learning: Discrete choice experiment.** Next, we sought to examine preferences for specific characteristics of the water delivery service and to identify any impact of experiential learning on preferences for these product characteristics. To achieve this aim, we conducted a *discrete choice experiment (DCE)* using a pre-defined set of product characteristics identified during formative interviews and a review of the literature. Briefly, DCEs are designed to elicit stated preferences for specific product characteristics by providing a respondent with a choice of two or more hypothetical products. Each product contains a different set of the characteristics of interest. The respondent considers the two products and then chooses one or the other based on their implicit preference for the given combination of characteristics contained in that product. This procedure is then repeated several times, each time with a new set of products containing a different combination of those characteristics. After repeated selections, multivariate regression analysis allows the researcher to determine if any specific product characteristics played a stronger role in the decision to select a given product option. To conduct this procedure, respondents were first sampled during the baseline survey (before social marketing exercises and the random price auction, thus before they were given any new information about the water delivery service). Respondents were sampled again during endline surveys. The goal was to identify their preference for a hypothetical product comprised of a randomly generated set of product characteristics versus a status quo product meant to mimic their current water supply. This procedure follows methods explored in detail in the literature [39–43].

The most important product characteristics identified from formative interviews and published literature included price, taste, convenience, safety, temperature and whether one's neighbors used the same product [4, 7, 37]. First, we took each of these characteristics (following Bridges et al. [40]) and converted them into discrete variables as displayed in **Table 2**: For price, we use a categorical variable with standard interval values of ₹0, ₹3, ₹6 and ₹9; for taste (*t*), we categorized into (0 = tastes like iron; 1 = tastes iron-free); for convenience (*d*), we categorized into (0 = on demand; 1 = call for delivery); for safety (*h*) we categorized into (0 = may cause sickness for me or my children; 1 = will not cause sickness for me or my children); for temperature (*c*) we categorized into (0 = cold; 1 = warm), and for neighbors (*n*), we

**Table 2. Product characteristic attributes and level for discrete choice experiment.**

| Characteristic | Levels |
|---|---|
| i) Price | ₹0 - "Water that is free" |
| | ₹3 - "Water that costs ₹3" |
| | ₹6 - "Water that costs ₹6" |
| | ₹9 - "Water that costs ₹9" |
| ii) Taste | (0) "Tastes like iron" |
| | (1) "Tastes nice" |
| iii) Convenience | (0) "Must be ordered for delivery" |
| | (1) "You can get whenever you like" |
| iv) Health | (0) "May not be safe to use" |
| | (1) "Is safe to use" |
| v) Temperature | (0) "Is room temperature" |
| | (1) "Is cold and refreshing" |
| vi) Neighbors | (0) "IS NOT used by most all of my neighbors" |
| | (1) "IS used by most all of my neighbors" |

Note: The price attribute has 4 levels while all other attributes have only two, meaning the universe of possible choice sets is 4 x 2 x 2 x 2 x 2 x 2 = 128 (possible combinations).

categorized into (0 = my neighbors use the same water source as I do; 1 = my neighbors use a different water source than I do).

Next, we generated a set of each of the product characteristic combinations, varying characteristics until arriving at 128 possible scenarios. Rather than compare each possible combination of scenarios (i.e., 128*127 = 16,256), we followed the example of Ryan and Ferrar [39] and created a single 'status-quo' scenario against which a randomly generated alternative was compared. Specifically, households were asked to imagine that their current water supply exhibited the qualities of the status quo scenario: a 'theoretical' hand-pump water source that has the following qualities: it is "i) free, ii) tastes like iron, iii) is retrieved on demand, iv) may make me or my children sick, v) cold, and vi) also used by most of my neighbors." This combination of characteristics was chosen to mimic the characteristics of pump water used by all the households in our sample (though some categories such as taste and safety could vary slightly from pump to pump). Following Johnson et al. [41], we eliminated all 'implausible' alternative product characteristic alternative scenarios that were strongly dominated by the status quo choice (e.g., "a delivered water product for a positive price that tastes bad, requires request for delivery, can make respondents sick, and is not used by neighbors") leaving a total of 124 possible alternatives to the status quo to be presented to 162 respondents (3 times each, totaling 486 choices) over two rounds. Although this is a large number of alternatives relative to the sample size, the power implications of this are mitigated by following the standard methods in this literature of estimating only a small number of choice-set parameters (eight). Our resulting confidence intervals confirm that we had sufficient power to estimate meaningful effect sizes.

Fig 4 provides an illustration of one of the 124 possible random choice sets faced by respondents with the status quo scenario in the left panel of the figure, and a randomly generated alternative set of characteristics in the right panel. Each respondent was asked in each baseline (3 random draws) and endline (3 new random draws) survey to choose between the same status quo scenario (left panel) and a new randomly generated comparison scenario (right panel) from the set of 124 non-dominated alternatives (in total, respondents faced 6 unique comparisons). Respondents could also elect to express "no preference" between alternatives, or "do not

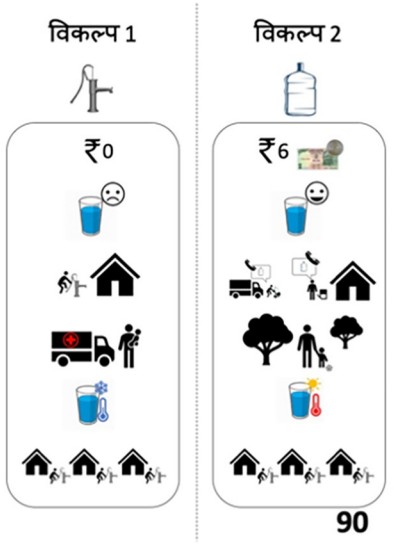

**Fig 4.** Example DCE choice set, status quo (left) random alternative (center) with scripted description of the choice set faced by respondents (right).

understand." Every randomly selected choice contained the same set of images on the left for the status quo condition, with the picture of the pump at the top, and a randomly generated alternative set on the right, with a picture of a bottle at the top (see **Fig 4**, right panel). In each case, enumerators read aloud the difference between each characteristic for the whole of each hypothetical product (or stated if they were the same). Respondents were given the chance to ask clarifying questions, and then asked to select either the status quo or the randomly generated alternative. The result of this process was a set of binary variables, one for each choice possible within each of the *i–vi* characteristics above, with a subset of 3 choice-set observations for each household at baseline and 3 additional observations at endline.

**2.4.1 Preferences and learning research questions.** We seek to answer two main research questions using the DCE described above, the first of which has two parts. First, regarding characteristic preferences we ask, *for which product characteristics do respondents express the strongest preference when choosing between the status quo (pump water) and the alternative (water delivery)*? As a concomitant question about learning, we also ask *whether these expressed preferences change over time*–from baseline (before the social marketing exercise) to endline surveys (after social marketing, price auctions and one week of water deliveries among households who choose to become customers). Our objective was to generate individual hypothesis tests for each characteristic to see, *ex ante*, whether preferences for treated water characteristics are the same as other household water treatment products (HWTP), and *ex post*, whether these preferences change in any ways not shown in the HWTP literature.

To answer these questions preferences for policy alternatives selected in our discrete choice experiment are modeled based on the Random Utility Model [44]. Econometrically, we analyze the results of this experiment using two equations in three models–at baseline (model 1), at endline (model 2) and pooled (model 3). We analyze our first two models using the following multivariate logistic regression (Eq 1):

$$g(\mu)_{hk} = \alpha + \beta_1 \boldsymbol{X}_{hk} + \epsilon_{hk} \tag{2}$$

where, for household $h$ and choice $k$, $g(\mu)_{hk}$ is the log-odds of taking water delivery over the status quo (the 'theoretical' household pump), $\alpha$ is the intercept, $X_{hk}$ is a vector of the product

characteristics (including price, taste, convenience, safety, temperature, and whether neighbors use the same or a different water source) and $\epsilon_{hk}$ is an error term clustered at the household level.

In addition to examining product characteristic preferences at baseline and endline using Eq 2, we also examine the change in these preferences for the vector $X_{hk}$ of characteristics over time $t$ using Eq 3, a pooled analysis with the addition of a dummy-variable for time ($time_t$) and interaction terms between each characteristic and time ($X_{hkt}*time_t$) to look for significant changes in characteristic preferences between survey rounds.

$$g(\mu)_{hkt} = \alpha + \beta_1 X_{hkt} + \beta_2 time_t + \beta_3 \boldsymbol{X_{hkt}}*time_t + \epsilon_{hkt} \tag{3}$$

Within the first two models, we examine the marginal output of each coefficient within the vector of product characteristics $X_{hk}$ for direction and statistical significance. Statistically significant coefficients at the 95%-level signify a failure to reject the corresponding hypothesis, that a preference is expressed by the respondent regarding that characteristic. The sign of the coefficient signifies the direction in which respondents value that characteristic against the status quo. In model 3 we examine the marginal output of the coefficients on the interaction terms ($\boldsymbol{X_{hkt}}*time_t$), the size and direction of which correspond to a percentage point change in preferences towards or away from that characteristic from baseline to endline. Statistically significant coefficients at the 95%-level for coefficients on any interaction terms signifies a significant change in characteristic preference from baseline to endline.

### 2.5. Stated preferences versus WTP: Research questions

Finally, we combine the results of our DCE with those of the random price auction to compare real auction WTP with stated preferences for the water delivery service. Households who both win the random price auction and subsequently choose to purchase the product (the eventual customers) are among those with the highest willingness to pay for the product and have the strongest revealed preference for the bottled water product by virtue of having chosen to purchase after winning. Thus, we ask: *How do preferences for individual product characteristics differ among customers (those with the highest willingness to pay) versus non-customers*? Regarding learning, we further ask: *How do these preferences change over time among customers versus non-customers*? To answer these questions, we examine the results of our DCE separately among only customers (a.k.a., a subset of those with the highest revealed willingness to pay) versus non-customers. Specifically, we repeat models 1–3 of the DCE (described previously) for the subset of customers only (n = 56) and separately for non-customers (n = 104). We also examine the mean difference in responses to opinion questions about the product between these two groups for another set of endline questions about product characteristics.

## 3. Results

We begin by examining the results on willingness to pay for the potable water delivery service in Auctions 1 and 2, followed by a comparison of the two auction results to examine any water subsidy effects on demand. Next, we explore product characteristic preferences from the DCE and endline surveys. Finally, we compare the revealed preference results from the random price auctions with stated preference results from our DCEs and endline questionnaires.

### 3.1 WTP for potable water

Table 3 shows mean values for baseline household-level characteristics for the study sample. Column 1 shows the mean values for each covariate for the whole sample. Columns 2 and 3 show the results of mean comparison tests for each of these covariates between those who

**Table 3. Baseline characteristics by auction group.**

| | Mean | (Auction 1) Delivery package auction | (Auction 2) Hardware-only auction | p-value | n |
|---|---|---|---|---|---|
| Total number of adults (≥14 years) | 3.3 | 3.3 | 3.2 | 0.961 | 162 |
| | | (0.2162) | (0.1787) | | |
| Total number of children (<14 years) | 2.2 | 2.0 | 2.4 | 0.140 | 162 |
| | | (0.1619) | (0.1947) | | |
| *Maithili* spoken at home | 0.222 | 0.275 | 0.183 | 0.163 | 162 |
| | | (0.0542) | (0.0403) | | |
| *Theti* spoken at home | 0.765 | 0.710 | 0.806 | 0.154 | 162 |
| | | (0.0550) | (0.0412) | | |
| Respondent is head of household | 0.321 | 0.406 | 0.258 | 0.047** | 162 |
| | | (0.0595) | (0.0456) | | |
| . . . typically makes purchase decisions | 0.759 | 0.826 | 0.710 | 0.088* | 162 |
| | | (0.0460) | (0.0473) | | |
| Female head has any formal education | 0.155 | 0.217 | 0.109 | 0.060* | 161 |
| | | (0.0500) | (0.0326) | | |
| Female head some secondary education | 0.093 | 0.101 | 0.087 | 0.756 | 161 |
| | | (0.0366) | (0.0295) | | |
| Male head has any formal education | 0.519 | 0.612 | 0.449 | 0.045** | 156 |
| | | (0.0600) | (0.0530) | | |
| Male head some secondary education | 0.301 | 0.403 | 0.225 | 0.016** | 156 |
| | | (0.0604) | (0.0445) | | |
| Asset index | 0.315 | 0.325 | 0.308 | 0.363 | 162 |
| | | (0.0162) | (0.0110) | | |
| Reported monthly income | ₹10,658 | ₹12,166 | ₹9,527 | 0.065* | 161 |
| [Median = ₹9,000 all groups] | | (1497) | (500) | | |
| Main occupation is wage labor | 0.667 | 0.609 | 0.710 | 0.180 | 162 |
| | | (0.0592) | (0.0473) | | |
| Main occupation is salaried labor | 0.037 | 0.058 | 0.022 | 0.227 | 162 |
| | | (0.0283) | (0.0151) | | |
| Main occupation is farming | 0.117 | 0.174 | 0.075 | 0.054* | 162 |
| | | (0.0460) | (0.0275) | | |
| Main occupation is self-employment | 0.173 | 0.145 | 0.194 | 0.421 | 162 |
| | | (0.0427) | (0.0412) | | |
| Improved structure (roof, walls, floor) | 0.603 | 0.628 | 0.584 | 0.334 | 162 |
| | | (0.0327) | (0.0304) | | |
| Ever treat drinking water | 0.179 | 0.188 | 0.172 | 0.790 | 162 |
| | | (0.0474) | (0.0393) | | |
| Children <5 with diarrhea in last 2 weeks | 0.265 | 0.232 | 0.290 | 0.408 | 162 |
| | | (0.0512) | (0.0473) | | |
| Religion of household is Hindu | 0.778 | 0.928 | 0.667 | 0.000*** | 162 |
| | | (0.0314) | (0.0491) | | |
| n | 162 | 69 | 93 | | |

Notes: Standard errors *(in parentheses)*; T-tests compare households in either "hardware-only draw" group versus "delivery package draw" group; "Hardware-only draw" are households who only bid on price of hardware after being randomized into ₹35/week versus ₹70/week water price groups; "Delivery package draw" are households who bid on the combined price of hardware (bottle & dispenser) and one week of daily water delivery.

**Table 4. Results of random price auctions.**

| | Auction 1 (hardware + deliveries) | Auction 2 (hardware only) | |
| --- | --- | --- | --- |
| | | ₹35 water discount | no water discount[1] |
| SHRI retail price of good auctioned | ₹320 | ₹250 | ₹250 |
| Additional price faced for deliveries | (included) | ₹35 | ₹70 |
| *Among all bidders* | | | |
| Total bids (n) | 69 | 59 | 34 |
| Mean bid price | ₹164 | ₹138 | ₹55 |
| (s.d.) | (117) | (118) | (95) |
| Median bid price | ₹180 | ₹160 | ₹0 |
| *Among positive bidders* | | | |
| Total positive bids (n) | 51 | 37 | 10 |
| Mean bid price | ₹222 | ₹220 | ₹186 |
| (s.d.) | (73) | (62) | (75) |
| Median bid price | ₹240 | ₹240 | ₹180 |
| Total auction winners (n) | 32 | 31 | 3 |
| Initial vs. final bids (final round)[$] | | | |
| % with same initial and final bids | 60.9 | 81.7 | |
| Avg ₹ difference in all bids | ₹48 | ₹16 | |
| Avg ₹ difference in bids (if different) | ₹123 | ₹89 | |
| % of losers who try to purchase | 0.0 | 0.0 | |
| % of losers who regret low bid price | 5.3 | 3.9 | |

Notes: 1. The "no water discount" group thus had to incorporate the additional cost of one week of water, valued at ₹70, into their purchase decision; $ Tripwire questions were not collected consistently during the practice round, thus those outcomes are not available here. Auction 1 was conducted before Auction 2

played Auction 1 versus those who played Auction 2, respectively. As noted, these groups were not randomized into the delivery package auction versus the hardware-only auction. Thus, there are significant differences between groups, including in the proportion of respondents who were heads of household, educational achievement of female and male heads of household, self-reported income, main occupation, and religion.

**3.1.1 Results of random price auctions.** **Table 4** and **Fig 5** show the results of the two random price auctions. In **Fig 5**, the vertical axis shows the proportion of households bidding at or below a given price level. The horizonal axis shows the prices at which households bid for the water delivery service. The number of bids is represented by the size of the circles on the scatterplot, and the fitted trendlines show the willingness to pay for each product option. In the first auction (panels I and II), respondents bid on the combined package of hardware plus one week of water delivery. The normal market price for the delivery package provided by the NGO (bottle, dispenser and seven 20-litre water deliveries) is ₹320, denoted by the black vertical line. The panels on the left (I and IV) show all the bids placed by those who played the auction, while the right panels (II and III) show the bids of only those who "won" the respective auctions.

In Auction 1, respondents were asked to bid on a package of seven water deliveries plus requisite hardware. The average bid was ₹164 among all bidders (Panel I) and ₹222 among all non-zero bidders (Panel II). In Auction 2 (panels III and IV), respondents were first asked to draw a number from a hat with a 50:50 chance to receive a 50% discount on the price of 7 water deliveries. The dashed lines show the WTP among those who received the ₹35 discounts, while the dotted lines show the WTP among those who did not. In Panel IV, the average bid among winners was ₹138, and among non-winners was ₹55. In Panel III, the average (non-

**Table 5. OLS regression of mean price bid in random price auction by Auction 2 treatment status relative to Auction 1 (combined package).**

|  | (1) | | | (2) | | |
|---|---|---|---|---|---|---|
| VARIABLES | Coef. | (se) | 95% CI | Coef. | (se) | 95% CI |
| Auction 1 (hardware and delivery prices combined) | (reference) | | | (reference) | | |
| Auction 2 (given ₹70 for seven deliveries) | -109.6*** | (23.65) | -156.35, -62.93 | -86.8*** | (28.43) | -143.15, -30.46 |
| Auction 2 (given ₹35 for seven deliveries) | -26.4 | (20.01) | -65.91, 13.14 | -8.1 | (22.86) | -53.38, 37.21 |
| Respondent is household head |  |  |  | -2.1 | (20.95) | -43.61, 39.41 |
| Male household head has any education |  |  |  | 8.6 | (21.25) | -33.53, 50.69 |
| Main household religion = Hinduism |  |  |  | -27.6 | (34.52) | -96.05, 40.78 |
| Male head has some secondary education |  |  |  | 5.1 | (29.48) | -53.32, 63.52 |
| Female head has any education |  |  |  | 14.0 | (26.62) | -38.71, 66.78 |
| Asset index (0–1) |  |  |  | -184.0** | (90.63) | -363.57, -4.34 |
| Stated monthly income (/₹1000) |  |  |  | 2.7*** | (0.80) | 1.13, 4.31 |
| Farming/agriculture is primary occupation |  |  |  | 44.7 | (27.88) | -10.58, 99.92 |
| Respondent typically makes purchase decisions |  |  |  | 57.4*** | (20.52) | 16.74, 98.07 |
| Neighborhood fixed effects | No | | | Yes | | |
| Constant | 164.4*** | (13.59) | 137.51, 191.18 | 100.9* | (53.97) | -6.08, 207.84 |
| Observations | 162 | | | 155 | | |
| R-squared | 0.1205 | | | 0.2825 | | |

Notes: Robust standard errors in parentheses, clustered at the structure-level; *** p<0.01, ** p<0.05, * p<0.1; Main regression coefficients are for treatment ("Auction 2 (given ₹70 for seven deliveries)") and control ("Auction 2 (given ₹35 for seven deliveries)") groups relative to the prices bid in "Auction 1 (hardware and delivery prices combined)". Negative coefficients under either treatment or control condition in Auction 2 signifies the average bid price under that condition are lower than the total price bid in Auction 1.

zero) bid among winners was ₹220 and the average non-zero bid among non-winners was ₹186. Additional results for both auctions can again be found in Table 4, including exploring whether initial bids matched final bids along with the amounts of bid difference, as well as any feelings of regret expressed by those who lost the auction. Results show between 61%-82% agreement in bids between rounds, with low levels of expressed regret among auction losers. Anecdotally, study respondents were highly engaged and excited to play the auction "game."

Because the WTP auctions represent a lab-in-the-field experiment, there were some violations to the rules of the game for which we could not control. Out of the 66 winners from either auction, a total of 10 respondents– 6 in Auction 1 and 4 in Auction 2 –later refused to pay for the hardware and delivery service and recanted on the rules of the game before deliveries could commence (all those who recanted in Auction 2 had randomly drawn a 50% discount on water deliveries). Because this was a vulnerable population, we made exceptions for these families to recoup the money they had agreed to spend. But we did not advertise this practice. The results we show in Tables 4 and 5 and in Figs 5 and 6 include the original bids from these 10 respondents who later recanted. When we later examine customer preferences starting in section 3.2, these 10 winners are counted among "non-customers."

**3.1.2 Comparison of Auction 1 vs 2: Impact of subsidies on demand for water delivery.** In Fig 6, we compare auction results, adjusting for the separate price of water delivery by combining the randomly drawn water delivery price with each hardware price bid by households in Auction 2. These plots are overlaid on the unadjusted results from Auction 1 for comparison and suggest that the average WTP for the whole water delivery service among those who did not receive a discount in Auction 2 was lower than among those who did receive a discount or those in Auction 1. Table 5 examines the results shown in Fig 6, controlling for baseline covariate imbalance between the two auctions. In this regression, we create auction group

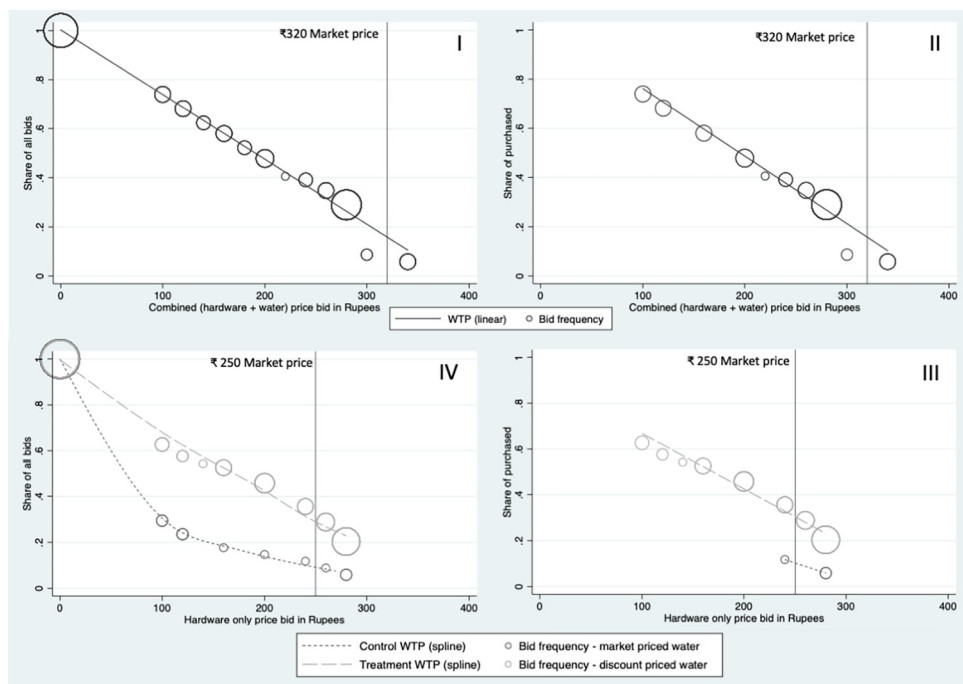

**Fig 5.** WTP for hardware and 7 deliveries (top) and hardware only (bottom) among all bidders (left) and only auction "winners" (right).

dummy variables (Auction 1; Auction 2, water discount; and Auction 2, no water discount) to regress on average bid price using OLS. In each model presented, bids for the two randomly generated water delivery prices (discounted at ₹35 versus no discount at ₹70) in the second auction are compared to the reference group bids from Auction 1. In model 1, before adjustment, we find that receiving no discount on water price in Auction 2 was associated with a ₹110 lower bid price on the bottle and dispenser than the average bid in Auction 1 (p<0.01). After adjustment we find an average bid price ₹87 lower than the bids in Auction 1 (p<0.01). In both models, we find that receiving a discount had no effect on the overall bid price among winners, suggesting that an unexpected, small promotional discount was not effective at increasing demand for the overall service.

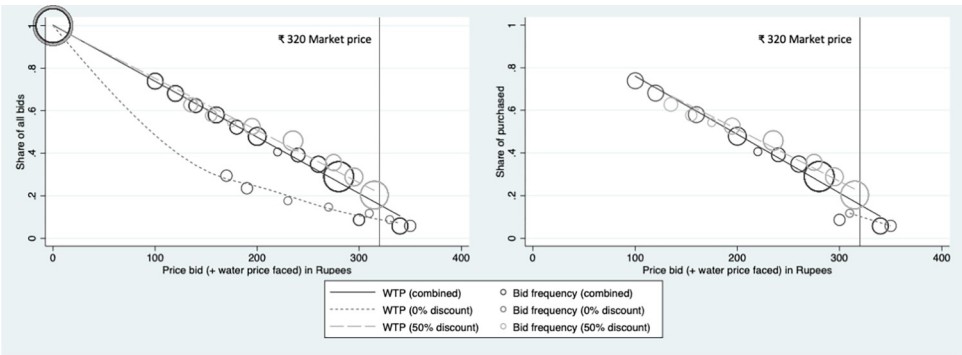

**Fig 6.** Willingness to pay for hardware and seven water deliveries among all bidders (left) and among winners who purchased (right)–Auction 2 participants scaled to include delivery cost drawn.

## 3.2 Customer preferences

The DCE procedure was conducted among all 162 households in the study sample during baseline interviews (before social marketing and the random price auction took place) and again at endline among 160 households that were not lost to follow-up. Each respondent was asked to choose between the status quo (handpump) and a randomly selected alternative (delivery) three times (3*162) for a total of 486 selections at baseline and (3*160) 480 selections at endline. Our approach was non-standard and did not use DCE software to achieve attribute balance and D-efficiency. Nonetheless, attribute levels were well balanced in each of the two surveys, as can be found in **Appendix Table 1 in S1 File**. All but two of the 124 possible non-dominated alternative scenarios were randomly presented during the baseline draw, while all other alternatives were presented at least once. At baseline, responses were as follows: 'alternative' (308), 'status quo' (150), 'no preference' (27) and 'do not understand' (1)–this respondent was dropped. A total of 59 households (36.4%) chose the alternative scenario in all three draws, 21 households (13%) always chose the status quo, one household chose 'no preference' all three times, and the remaining 81 households (50%) varied their responses. At follow-up, a total of three possible alternative scenarios were not generated. The total number of possible endline responses were as follows: 'alternative' (299), 'status quo' (176), 'no preference' (5), 'do not understand' (0). A total of 70 households (43.8%) always chose the alternative, 34 (21.3%) always chose the status quo, and the remaining 56 (35.0%) varied their responses.

**Table 6** shows the results of the DCE. In Model 1, the marginal results of the logistic regression at baseline show that the largest significant coefficients are on delivery price and product safety. For delivery price, we find that, compared to a price of 0, respondents are 14.9 percentage points less likely to accept a price of ₹3 per bottle (p<0.05), 18.9 percentage points less likely to accept a price of ₹6 per bottle (p<0.01), and 15.5 percentage points less likely to accept a per bottle price of ₹9 (p<0.05). We also find that respondents are 10.1 percentage points more likely to choose a product that is not likely to make them or their children sick (p<0.05). At follow-up (model 2), we find that preferences for delivery price are somewhat attenuated while coefficients on taste and convenience have changed in magnitude and significance. Respondents are 10.1 percentage points more likely to prefer water that does not taste like iron (p<0.01) and 9.0 percentage points less likely to prefer a product that they must call ahead to attain (p<0.10). The magnitude, size, and direction of the coefficient on product safety remains mostly unchanged. In our pooled analysis in model 3, we find that interaction terms for taste and temperature are significant at the 90 percent level, suggesting that the most significant shifts in preference over the two time periods were: a) away from water with an iron taste, and b) towards water that was cold.

## 3.3 Comparison of revealed to stated preferences

Finally, we examine the results of our analyses of stated preferences (both the DCE and the stated preferences at endline) with the results of the random price auctions. Specifically, among 162 participating in the auctions, 98 (60.5%) bid a positive (non-zero) price for the delivery service and 56 purchased the product after winning the auction. This group of 56 eventual customers has the highest willingness to pay for the product and experienced the product through purchase of the hardware and 7 deliveries. We examine the difference in stated preferences between these *customers* (who did experience the product through purchasing water deliveries) and *non-customers* (who did not) below.

We start by examining the difference in DCE results by customer status. **Table 7** shows DCE results as the marginal output of a multivariate logistic regression among customers only. We find that, at baseline (model 1) among customers with the highest revealed WTP, the three

**Table 6. Discrete choice experiment–multivariate logistic regression of decision on product characteristics (dy/dx).**

| VARIABLES | (1) Baseline preferences | | | (2) Endline preferences | | | (3) Pooled analysis | | |
|---|---|---|---|---|---|---|---|---|---|
| | dy/dx | (se) | 95% CI | dy/dx | (se) | 95% CI | dy/dx | (se) | 95% CI |
| ₹0/delivery price | (reference) | | | (reference) | | | (reference) | | |
| ₹3/delivery price | -0.148** | (0.0655) | -0.2765, -0.0198 | -0.063 | (0.0619) | -0.1845, 0.0582 | -0.153** | (0.0667) | -0.2836, -0.0220 |
| ₹6/delivery price | -0.189*** | (0.0653) | -0.3167, -0.0608 | -0.097* | (0.0585) | -0.2117, 0.0174 | -0.194*** | (0.0658) | -0.3224, -0.0645 |
| ₹9/delivery price | -0.151** | (0.0593) | -0.2670, -0.0345 | -0.211*** | (0.0666) | -0.3419, -0.0808 | -0.155** | (0.0616) | -0.2761, -0.0346 |
| Taste | 0.004 | (0.0451) | -0.0844, 0.0926 | 0.108*** | (0.0406) | 0.0289, 0.1880 | 0.004 | (0.0460) | -0.0861, 0.0945 |
| Convenience | -0.029 | (0.0472) | -0.1210, 0.0640 | -0.090* | (0.0469) | -0.1816, 0.0022 | -0.029 | (0.0480) | -0.1233, 0.0651 |
| Safety | 0.101** | (0.0445) | 0.0140, 0.1883 | 0.089** | (0.0441) | 0.0021, 0.1748 | 0.103** | (0.0460) | 0.0132, 0.1933 |
| Temperature | 0.056 | (0.0426) | -0.0280, 0.1389 | -0.059 | (0.0448) | -0.1473, 0.0285 | 0.057 | (0.0434) | -0.0285, 0.1417 |
| Neighbors use | 0.054 | (0.0451) | -0.0344, 0.1424 | -0.018 | (0.0439) | -0.1035, 0.0684 | 0.055 | (0.0463) | -0.0356, 0.1459 |
| Time (0 = baseline, 1 = endline) | | | | | | | -0.250** | (0.1075) | -0.4602, -0.0388 |
| CHARACTERISTIC*TIME | | | | | | | | | |
| ₹0/delivery price * Time | | | | | | | (reference) | | |
| ₹3/delivery price * Time | | | | | | | 0.095 | (0.0817) | -0.0655, 0.2547 |
| ₹6/delivery price * Time | | | | | | | 0.099 | (0.0871) | -0.0714, 0.2699 |
| ₹9/delivery price * Time | | | | | | | -0.037 | (0.1053) | -0.2437, 0.1692 |
| Taste * Time | | | | | | | 0.102* | (0.0585) | -0.0126, 0.2169 |
| Convenience * Time | | | | | | | -0.059 | (0.0644) | -0.1850, 0.0672 |
| Safety * Time | | | | | | | -0.016 | (0.0616) | -0.1372, 0.1044 |
| Temperature * Time | | | | | | | -0.115* | (0.0661) | -0.2445, 0.0147 |
| Neighbors use * Time | | | | | | | -0.072 | (0.0640) | -0.1978, 0.0531 |
| Pseudo R² | 0.0323 | | | 0.0439 | | | 0.0398 | | |
| Household observations | 161 | | | 160 | | | 162 | | |
| Choice observations | 458 | | | 475 | | | 933 | | |

Notes: Status Quo = 0, Alternative = 1; Alternative-specific constants are suppressed in the table above; "dy/dx" denotes marginal effects; *** $p<0.01$, ** $p<0.05$, * $p<0.1$; Robust standard errors clustered at household-level; Comparison of characteristics to status quo scenario: "delivery price" compares to '₹0 delivery price' reference; "Taste" compares to a "Iron taste = 0"; "Convenience" compares to an "on-demand = 0" comparison; "Safety" compares to a "might cause sickness = 0" comparison; "Temperature" compares to a "cold = 0" comparison; "Neighbors use" compares to a "Neighbors use same source = 0" comparison.

coefficients on delivery price are not statistically significant. Eventual customers are 12.5 percent more likely to purchase the alternative product despite having to call ahead for the delivery (p<0.1). At endline (model 2) we find that customers no longer express a preference for product convenience. Meanwhile, customers at endline were 11.9 percent more likely to purchase an alternative that tasted better than the status quo handpump water (p<0.05). In model 3 we find significant negative changes in preference for convenience and whether neighbors use the same product.

Table 8 shows DCE results among non-customers only. At baseline, the three coefficients on delivery price suggest that, compared to zero-price status quo water, non-customers were 17.5 percentage points less likely to purchase alternative water at ₹3 (p<0.1), 32.5 percentage points less likely at ₹6 (p<0.01) and 17.7 percentage points less likely at ₹9 (p<0.05). Additionally, non-customers were 10.1 and 9.5 percentage points more likely to choose an alternative to the status quo for water that was safe (p<0.1) and warm (p<0.1), respectively. At endline, after experiencing only the taste test and social marketing (and possibly seeing neighbor customers spend a week consuming the product), non-customers remained significantly less likely to prefer an alternative over a free status quo at any price. Meanwhile, non-

**Table 7. DCE, multivariate logistic regression of decision on product characteristics (dy/dx)—customer households only.**

| VARIABLES | (1) Baseline DCE | | | (2) Endline DCE | | | (3) Pooled DCE | | |
|---|---|---|---|---|---|---|---|---|---|
| | dy/dx | (se) | 95% CI | dy/dx | (se) | 95% CI | dy/dx | (se) | 95% CI |
| ₹0/delivery price | (reference) | | | (reference) | | | (reference) | | |
| ₹3/delivery price | -0.043 | (0.0800) | -0.1994, 0.1144 | 0.095 | (0.0858) | -0.0732, 0.2632 | -0.044 | (0.0828) | -0.2058, 0.1186 |
| ₹6/delivery price | 0.037 | (0.0666) | -0.0933, 0.1677 | -0.035 | (0.0942) | -0.2193, 0.1501 | 0.039 | (0.0688) | -0.0960, 0.1736 |
| ₹9/delivery price | -0.125 | (0.0760) | -0.2733, 0.0244 | 0.007 | (0.0953) | -0.1794, 0.1943 | -0.126 | (0.0780) | -0.2790, 0.0269 |
| Taste | 0.013 | (0.0627) | -0.1100, 0.1358 | 0.119** | (0.0523) | 0.0161, 0.2212 | 0.013 | (0.0624) | -0.1093, 0.1352 |
| Convenience | 0.125* | (0.0677) | -0.0080, 0.2573 | -0.022 | (0.0564) | -0.1328, 0.0882 | 0.125* | (0.0707) | -0.0141, 0.2632 |
| Safety | 0.094 | (0.0690) | -0.0408, 0.2297 | 0.051 | (0.0549) | -0.0567, 0.1585 | 0.094 | (0.0716) | -0.0460, 0.2347 |
| Temperature | -0.026 | (0.0596) | -0.1424, 0.0911 | -0.030 | (0.0581) | -0.1434, 0.0842 | -0.026 | (0.0597) | -0.1427, 0.0915 |
| Neighbors use | 0.098 | (0.0663) | -0.0322, 0.2275 | -0.067 | (0.0660) | -0.1961, 0.0626 | 0.098 | (0.0683) | -0.0363, 0.2314 |
| Time (0 = baseline, 1 = endline) | | | | | | | -0.240 | (0.1613) | -0.5562, 0.0762 |
| INTERACTION TERMS (CHARACTERISTIC*TIME) | | | | | | | | | |
| ₹0/delivery price * Time | | | | | | | (reference) | | |
| ₹3/delivery price * Time | | | | | | | 0.125 | (0.0888) | -0.0493, 0.2989 |
| ₹6/delivery price * Time | | | | | | | -0.096 | (0.1852) | -0.4590, 0.2670 |
| ₹9/delivery price * Time | | | | | | | 0.105 | (0.1007) | -0.0929, 0.3019 |
| Taste * Time | | | | | | | 0.106 | (0.0809) | -0.0526, 0.2643 |
| Convenience * Time | | | | | | | -0.147* | (0.0812) | -0.3060, 0.0123 |
| Safety * Time | | | | | | | -0.043 | (0.1054) | -0.2499, 0.1632 |
| Temperature * Time | | | | | | | -0.004 | (0.0887) | -0.1779, 0.1699 |
| Neighbors use * Time | | | | | | | -0.164* | (0.0982) | -0.3569, 0.0281 |
| Pseudo R$^2$ | 0.0666 | | | 0.0557 | | | 0.0613 | | |
| Household observations | 56 | | | 56 | | | 56 | | |
| Choice observations | 166 | | | 164 | | | 330 | | |

Notes: Status Quo = 0, Alternate = 1; Alternative-specific constants are suppressed in the table above; "dy/dx" denotes marginal effects; *** p<0.01, ** p<0.05, * p<0.1; Robust standard errors clustered at household-level; Comparison of characteristics to status quo scenario: "delivery price" compares to '₹0 delivery price' reference; "Taste" compares to a "Iron taste = 0"; "Convenience" compares to an "on-demand = 0" comparison; "Safety" compares to a "might cause sickness = 0" comparison; "Temperature" compares to a "cold = 0" comparison; "Neighbors use" compares to a "Neighbors use same source = 0" comparison.

customers were 10.6 percentage points more likely to choose an alternative to the status quo that tasted good (p<0.05), 10.8 percentage points less likely to choose an alternative for which they would need to call ahead (p<0.10), and 10.5 percentage points more likely to choose an alternative that was safe to drink (p<0.10).

In addition to the DCE, we also asked respondents to express their opinions about several product characteristics during the follow up survey, including those not used in the DCE. Respondent preferences were stated as either "I like the following. . ." or "I dislike the following. . .". We find that among the product characteristics most "liked" by respondents, over 52% report liking the taste of the water, 32% the convenience of the bottle and dispenser, 28% the convenience of delivery, and 24% the safety of the water. The most disliked characteristics of the delivery service were hardware price (41%), water price (39%), water temperature (29%), and water taste (18%). **Table 9** shows the results of this set of opinion questions compared between customers and non-customers. Mean comparisons for each characteristic preference were conducted between these two groups. Compared to customers, significantly more non-customers expressed disliking the price of the water (p<0.01) and the price of the hardware (p<0.01). Meanwhile, compared to non-customers, customers expressed a

**Table 8. DCE, multivariate logistic regression of decision on product characteristics (dy/dx)–non-customer households only.**

| | (1) | | | (2) | | | (3) | | |
|---|---|---|---|---|---|---|---|---|---|
| | Baseline DCE | | | Endline DCE | | | Pooled DCE | | |
| VARIABLES | dy/dx | (se) | 95% CI | dy/dx | (se) | 95% CI | dy/dx | (se) | 95% CI |
| ₹0/delivery price | (reference) | | | (reference) | | | (reference) | | |
| ₹3/delivery price | -0.175* | (0.0903) | -0.3520, 0.0019 | -0.143* | (0.0757) | -0.2910, 0.0055 | -0.177** | (0.0901) | -0.3532, -0.0002 |
| ₹6/delivery price | -0.325*** | (0.0867) | -0.4950, -0.1550 | -0.147** | (0.0717) | -0.2875, -0.0065 | -0.321*** | (0.0825) | -0.4823, -0.1589 |
| ₹9/delivery price | -0.177** | (0.0811) | -0.3361, -0.0184 | -0.335*** | (0.0810) | -0.4941, -0.1766 | -0.179** | (0.0828) | -0.3410, -0.0166 |
| Taste | 0.029 | (0.0610) | -0.0905, 0.1486 | 0.106** | (0.0524) | 0.0027, 0.2082 | 0.029 | (0.0613) | -0.0910, 0.1494 |
| Convenience | -0.085 | (0.0607) | -0.2043, 0.0336 | -0.108* | (0.0587) | -0.2226, 0.0076 | -0.086 | (0.0611) | -0.2058, 0.0339 |
| Safety | 0.101* | (0.0576) | -0.0119, 0.2139 | 0.105* | (0.0581) | -0.0084, 0.2192 | 0.102* | (0.0586) | -0.0131, 0.2165 |
| Temperature | 0.095* | (0.0560) | -0.0151, 0.2045 | -0.038 | (0.0585) | -0.1523, 0.0770 | 0.095* | (0.0565) | -0.0154, 0.2061 |
| Neighbors use | 0.014 | (0.0559) | -0.0959, 0.1234 | 0.034 | (0.0553) | -0.0740, 0.1428 | 0.014 | (0.0564) | -0.0966, 0.1243 |
| Time (0 = baseline, 1 = endline) | | | | | | | -0.195 | (0.1352) | -0.4595, 0.0705 |
| INTERACTION TERMS (CHARACTERISTIC*TIME) | | | | | | | | | |
| ₹0/delivery price * Time | | | | | | | (reference) | | |
| ₹3/delivery price * Time | | | | | | | 0.045 | (0.1113) | -0.1734, 0.2631 |
| ₹6/delivery price * Time | | | | | | | 0.169* | (0.1029) | -0.0323, 0.3709 |
| ₹9/delivery price * Time | | | | | | | -0.139 | (0.1287) | -0.3909, 0.1136 |
| Taste * Time | | | | | | | 0.076 | (0.0730) | -0.0676, 0.2187 |
| Convenience * Time | | | | | | | -0.021 | (0.0836) | -0.1847, 0.1429 |
| Safety * Time | | | | | | | 0.003 | (0.0742) | -0.1424, 0.1486 |
| Temperature * Time | | | | | | | -0.133 | (0.0879) | -0.3051, 0.0395 |
| Neighbors use * Time | | | | | | | 0.020 | (0.0768) | -0.1301, 0.1708 |
| Pseudo R² | 0.0572 | | | 0.0686 | | | 0.0657 | | |
| Household observations | 103 | | | 104 | | | 104 | | |
| Choice observations | 286 | | | 311 | | | 597 | | |

Notes: Status Quo = 0, Alternative = 1; Alternative-specific constants are suppressed in the table above; "dy/dx" denotes marginal effects; *** p<0.01, ** p<0.05, * p<0.1; Robust standard errors clustered at household-level; Comparison of characteristics to status quo scenario: "delivery price" compares to '₹0 delivery price' reference; "Taste" compares to a "Iron taste = 0"; "Convenience" compares to an "on-demand = 0" comparison; "Safety" compares to a "might cause sickness = 0" comparison; "Temperature" compares to a "cold = 0" comparison; "Neighbors use" compares to a "Neighbors use same source = 0" comparison.

significantly greater preference for water taste (p<0.01) and the convenience of hardware use (p<0.01).

To summarize our findings, in Auction 1 we find that 74% of respondents bid a non-zero price and, compared to the market price of ₹320 for the hardware and first seven deliveries, mean WTP was about half (₹164) among all bidders, and almost two-thirds of market price (₹222) among those bidding a positive price. In Auction 2, we find that 63% of those who knew they would receive a 50% water delivery discount placed a positive bid on hardware, versus only 29% of those who knew they would not. Mean WTP for the hardware-only (normally ₹250) was also higher among those receiving a water discount (₹138) versus those who did not (₹55). Among those in Auction 2 with positive bids, mean WTP for hardware was closer to market price among those receiving a water discount (₹222) compared to those who did not (₹186). Indeed, in our regression analyses we find evidence that receiving water discount offers may raise mean WTP for hardware compared to what would be expected in the absence of a discount offer. However, our subsequent analysis suggests that this initial difference is misleading, and likely driven by a decrease in mean WTP for hardware among those who missed out on discounts in Auction 2. Our findings led to a failure to reject alternative

**Table 9. Product characteristic preferences at endline among customers vs. non-customers.**

| | Do you "dislike" any of the following characteristics? | | | Do you "like" any of the following characteristics? | | |
|---|---|---|---|---|---|---|
| CHARACTERISTIC | Non-customers | Customers | p-value | Non-customers | Customers | p-value |
| The dispenser | 0.038 | 0.018 | 0.478 | 0.087 | 0.089 | 0.954 |
| | (0.0189) | (0.0179) | | (0.0277) | (0.0385) | |
| The bottle | 0.067 | 0 | 0.047** | 0.087 | 0.161 | 0.159 |
| | (0.0247) | (0.000) | | (0.0277) | (0.0495) | |
| Popularity among neighbors | 0.019 | 0 | 0.299 | 0.058 | 0.018 | 0.243 |
| | (0.0135) | (0.000) | | (0.0230) | (0.0179) | |
| Safety | 0.048 | 0.089 | 0.307 | 0.240 | 0.232 | 0.908 |
| | (0.0211) | (0.0385) | | (0.0421) | (0.0569) | |
| Water temperature | 0.269 | 0.339 | 0.357 | 0.019 | 0.125 | 0.005*** |
| | (0.0437) | (0.0638) | | (0.0135) | (0.0446) | |
| Convenience of delivery | 0.019 | 0.143 | 0.002*** | 0.260 | 0.321 | 0.410 |
| | (0.0135) | (0.0472) | | (0.0432) | (0.0630) | |
| Convenience of use | 0.029 | 0 | 0.202 | 0.212 | 0.518 | 0.000*** |
| | (0.0165) | (0.000) | | (0.0402) | (0.0674) | |
| Taste | 0.212 | 0.125 | 0.177 | 0.365 | 0.821 | 0.000*** |
| | (0.0402) | (0.0446) | | (0.0474) | (0.0516) | |
| Hardware price | 0.529 | 0.179 | 0.000*** | 0.029 | 0.071 | 0.212 |
| | (0.0492) | (0.0516) | | (0.0165) | (0.0347) | |
| Water price | 0.481 | 0.232 | 0.002*** | 0.038 | 0.161 | 0.007*** |
| | (0.0492) | (0.0569) | | (0.0189) | (0.0495) | |
| n | 104 | 56 | | 104 | 56 | |

Notes: Standard errors *italicized* in parentheses; ***p<0.01, **p<0.05, *p<0.1; Each characteristic assessed for 104 customers and 56 non-customers.

hypothesis 2, suggesting that the effect of missing out on small discounts had modest but larger negative effects on demand. We caution that there may be unobservable sources of bias inherent to those who were sampled for Auction 2 based on our non-random household listing procedure. However, the direction of potential bias is unclear.

Next, among all respondents, preferences for product characteristics are strongest regarding delivery price (all positive prices are less desirable than free water) and safety (safe water is more desirable than potentially unsafe water) at baseline. At endline, respondents additionally expressed a preference for products that taste good and that are convenient to obtain. There was considerable heterogeneity in preferences among customers versus non-customers at baseline and endline, with customers placing higher priority on convenience (at baseline) and taste (at endline) while non-customers preferred free water over any positive price.

## 4. Discussion and conclusions

Our main findings reveal several important lessons regarding demand for water delivery products in rural Bihar. First, although we find that there *is* latent demand for fully treated and delivered water, this demand may not be easily increased using very small promotional subsidies for the service alone. Second, though the market for similar services already exists, potential customers have incomplete information, and preferences for specific product characteristics are subject to change both through promotion and experiential learning. Third, the price of hardware is a barrier to entry and requires careful consideration for firms that

wish to maximize return on investment. Finally, those with the highest WTP for treated water seemed to place more importance on the safety, taste, and convenience of the product as major factors in their purchase decision, providing a roadmap for future marketing efforts.

Though we cannot generalize beyond this small-n study, our results can be seen as hypothesis-generating for designing interventions for delivered water services. For example, mean WTP for hardware among those who received a 50% water delivery discount (equivalent to $0.49 USD) for a short period (roughly one week) suggests that subsidies for water delivery might have scope to increase initial WTP for the requisite hardware, relative only to those who do not receive a discount. Indeed, these gains may not be driven by overall demand for the service, but instead by a dampening effect on demand among those who missed out on the small discount–akin to a phenomenon that social psychologists have termed the *inaction inertia effect* [45–50]. Increasing the size or lengthening the time frame of this discount could lead to a higher overall share of individuals purchasing water delivery, and positively learning about product qualities during a trial period compared to the status quo. Positive learning could then lead to greater sustained use after subsidies expire than we might expect in the absence of subsidies–a hypothesis that we test elsewhere (see: Cameron and Dow 2021) [51]. These findings may also be relevant where local governments are trying to scale up new water services. However, given the pilot nature of this study, well-powered experimental research is needed to identify any causal effects of discount offers; indeed, previous safe water studies have also found mixed effects from product discounts [9].

Although an experimental test-retest literature that examine the stability of preferences and WTP using discrete choice methods is well-established [52, 53], to our knowledge, our stated preferences exercise is among the first examples of a DCE with the same sample conducted before and after the introduction of a water product in an LMIC setting. Our results suggest that before experiencing the free taste test, social marketing and (among customers) the water delivery service itself, the most important product characteristics among all respondents were price and product safety. One week later, the importance of price was slightly diminished. According to self-reported opinion questions, many more respondents reported liking the taste of the delivered water than they did the safety of the product at endline (among especially customers, but also to a lesser extent among non-customers). By contrast, the results of the DCE suggest that safety and taste were roughly equally weighted as important drivers of demand among non-customers at endline. Meanwhile, like the self-reported opinion results, customers expressed a much greater preference for taste than safety at endline. These results are compatible with previous research in which taste and convenience frequently dominated health or safety in (stated) user preferences. For example, Blum and colleagues [54] find that, although non-customers expressed a strong preference for drinking water that is safe for them and their children, this preference was insufficient to induce demand, even after a social marketing exercise. Taste appears to be a more powerful driver.

Our findings also contrast with existing research on demand for household and point-of-use water treatment products that might also be used as a stop-gap for municipal systems. For example, using random price auctions to examine demand for several products in Tanzania, Burt et al. [20] find that median WTP for disposable water treatments *PuR* and *Waterguard* was half and 1/3 of retail price, respectively. For more durable siphon and pot filters, median WTP was 7%, and 11% of retail, respectively. Comparatively, demand for the first week of water deliveries (including hardware) in Bihar was 51% of retail price. Burt and colleagues' findings also suggest that more expensive and durable products such as filters are more desirable than other HWTS. However, the up-front cost of filters is a substantial limitation (for example, the price of pot filters represented 26.1% of median monthly income in Tanzania at the time) and is likely to require subsidization. Although Tanzania and India are very

difference contexts, and pot filters are likely to be far more durable than a plastic bottle and dispenser, the up-front cost of hardware for the delivery service in India represents only 3.2% of median monthly income. Nonetheless, even relatively modest startup costs can be a barrier to entry, and any clean water interventions requiring initial capital investments by the consumer should consider providing a trial period where they can test out the hardware or service to increase the number of new customers [51, 55].

Unlike Luoto and colleagues [19], who find that two-months of trial experience with four point-of-use water treatment products in Bangladesh lead to statistically significantly lower WTP for three of four products, we were unable to examine changes in willingness to pay among those who experienced a trial period with the delivery service versus those who did not. Through our DCE we do find that preferences for price remain relatively stable in both groups over time, but due to the difficulty of modifying the DCE mid-study, these findings are only specific to the price of water alone, as pre-specified. However, our findings on changes in stated preferences for the delivery service between customers and non-customers reveal important insights. We found that those with the highest *a priori* WTP (the customers) were driven by the desire for convenient delivery at the outset and learned that the product was less convenient than they had hoped over the short trial period. They also learned that they preferred the taste of treated water. The results of our separate endline survey questions largely agree with our DCE findings on customer preferences for taste and convenience, confirming previous research in Bihar [7p12].

The importance of delivery convenience should not be understated, and the design of similar programs should put emphasis on implementation fidelity by ensuring that delivery is timely and uninterrupted for new customers. Indeed, many field experiments conclude that even nominal entry prices and/or convenience barriers can dramatically reduce uptake of preventative health products and services [56, 57]. Our findings on preferences suggest that shifting the marketing emphasis towards the relative convenience of the service, and ensuring that deliveries are reliable, may help to remove what some economists have called "small hassles" [58] that reduce demand. Combined with trial periods to experience the product and smooth the price shock of hardware procurement, similar programs are likely to have the most meaningful impact on recruitment–and possibly retention–of new customers.

Our study has some important limitations that must temper our conclusions. First, this study is of a small, underpowered convenience sample of households located in communities clustered around a local NGO that provides water and sanitation services to the greater area. Our respondents lived close to the main road, which is often a sign of higher economic status than the community at large; [59] thus, results have limited generalizability to those who are less socioeconomically advantaged as well as those not living directly on a main road with easy access to delivery. Second, the comparison of results between Auction 1 and Auction 2 are based on non-equivalent groups. Further research is needed to adequately identify any possible negative effects of missing out on subsidies that our findings cannot rule out. Third, the time frame for participation was relatively short and exposure to the product and delivery service was limited to only one week. Therefore, we cannot comment on whether observed demand for purchased water will continue indefinitely. Fourth, and perhaps most importantly, 20-liters of water per family per day is insufficient to meet international standards for minimum clean water needs [60]. Indeed, as the average household size of our study was 5.5, we can confidently say that this falls below the *Joint Monitoring Programme for Water Supply, Sanitation and Hygiene* (JMP) definition of access to safely managed water. Even if the demand we observe continues, therefore, this limited volume, purchased water approach cannot be a substitute for piped municipal water services. These systems remain either absent or are growing only slowly in much of the low-income world.

## Supporting information

**S1 File. This file contains willingness to pay auction scripts and all appendix tables.**
(DOCX)

## Acknowledgments

The authors would like to acknowledge, without implicating, Lia CH Fernald and Paul J Gertler for their critical contributions to this work. Thanks also to Anoop Jain, Chandan Kumar, Prabin Kumar, Ranjeet Kumar, Kundan Kumar, Manoj Kumar and Vivek Kumar for their willingness to share in their valuable work at Sanitation Health Rights for India. We would also like to thank Golap Manjari Rout and Rajeev Kumar Mahto for exceptional field coordination as well as Roshan Kumar, Sailej Kumar, Shiv Kumar, Vikesh Kumar and Sumit Choudhary. Additional thanks to Claire Boone, Calvin Chiu, David Contreras-Loya, Maria Dieci, Harlan Downs-Tepper, Ada T. Kwan, Zachary Olson, Nicholas Otis, Nicole Perales, Helen Pitchik, Deepak Premkumar, Sarah Reynolds, Hector Rodriguez, Alexis Shenfil Smart and Zachary Wagner for valuable feedback on various iterations of this research. Any errors are our own.

## Author Contributions

**Conceptualization:** Drew B. Cameron, Isha Ray, William H. Dow.

**Data curation:** Drew B. Cameron.

**Formal analysis:** Drew B. Cameron.

**Funding acquisition:** Drew B. Cameron.

**Investigation:** Drew B. Cameron, Manoj Parida.

**Methodology:** Drew B. Cameron, Isha Ray, William H. Dow.

**Project administration:** Drew B. Cameron, Manoj Parida.

**Resources:** Drew B. Cameron, Manoj Parida.

**Software:** Drew B. Cameron, Manoj Parida.

**Supervision:** Drew B. Cameron, William H. Dow.

**Validation:** Drew B. Cameron, Manoj Parida.

**Visualization:** Drew B. Cameron.

**Writing – original draft:** Drew B. Cameron, Isha Ray.

**Writing – review & editing:** Drew B. Cameron, Isha Ray, William H. Dow.

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
