## [Decision Letter · Decision Letter 0]

11 Aug 2022

PONE-D-22-05048Product preferences and willingness to pay for potable water delivery: Experimental evidence from rural Bihar, IndiaPLOS ONE

Dear Dr. Cameron,

Thank you for submitting your manuscript to PLOS ONE. After careful consideration, we feel that it has merit but does not fully meet PLOS ONE’s publication criteria as it currently stands. Therefore, we invite you to submit a revised version of the manuscript that addresses the points raised during the review process.

We look forward to receiving your revised manuscript.

Kind regards,

Ilke Onur, Ph.D.

Academic Editor

PLOS ONE

Journal Requirements:

4. Thank you for stating the following financial disclosure: "DBC; IND-19059; the International Growth Centre; https://www.theigc.org; The funders had no role in study design, data collection and analysis, decision to publish, or preparation of the manuscript.

DBC; Center for Global Public Health, University of California, Berkeley, School of Public Health; https://cgph.berkeley.edu; The funders had no role in study design, data collection and analysis, decision to publish, or preparation of the manuscript."

We note that one or more of the authors is affiliated with the funding organization, indicating the funder may have had some role in the design, data collection, analysis or preparation of your manuscript for publication; in other words, the funder played an indirect role through the participation of the co-authors. If the funding organization did not play a role in the study design, data collection and analysis, decision to publish, or preparation of the manuscript and only provided financial support in the form of authors' salaries and/or research materials, please do the following:

a. Review your statements relating to the author contributions, and ensure you have specifically and accurately indicated the role(s) that these authors had in your study. These amendments should be made in the online form.

b. Confirm in your cover letter that you agree with the following statement, and we will change the online submission form on your behalf: 

“The funder provided support in the form of salaries for authors [insert relevant initials], but did not have any additional role in the study design, data collection and analysis, decision to publish, or preparation of the manuscript. The specific roles of these authors are articulated in the ‘author contributions’ section.

7. Please note that in order to use the direct billing option the corresponding author must be affiliated with the chosen institute. Please either amend your manuscript to change the affiliation or corresponding author, or email us at plosone@plos.org with a request to remove this option.

8. We note that you have referenced "Kremer M, Miguel EM, Mullainathan S, Null C, Zwane AP. Social Engineering: Evidence from a Suite of Take-up Experiments in Kenya. Unpublished Working Paper. 2011." which has currently not yet been accepted for publication. Please remove this from your References and amend this to state in the body of your manuscript: "Kremer M, Miguel EM, Mullainathan S, Null C, Zwane AP. Social Engineering: Evidence from a Suite of Take-up Experiments in Kenya. Unpublished Working Paper. 2011." as detailed online in our guide for authors

9. We note you have included a table to which you do not refer in the text of your manuscript. Please ensure that you refer to Table 4 in your text; if accepted, production will need this reference to link the reader to the Table.

Reviewers' comments:

Reviewer's Responses to Questions

**Comments to the Author**

1. Is the manuscript technically sound, and do the data support the conclusions?

Reviewer #1: Yes

Reviewer #2: Yes

2. Has the statistical analysis been performed appropriately and rigorously? 

Reviewer #1: Yes

Reviewer #2: Yes

3. Have the authors made all data underlying the findings in their manuscript fully available?

Reviewer #1: Yes

Reviewer #2: Yes

4. Is the manuscript presented in an intelligible fashion and written in standard English?

Reviewer #1: Yes

Reviewer #2: Yes

5. Review Comments to the Author

Reviewer #1: Review of “Product preferences and willingness to pay for potable water delivery: Experimental evidence from rural Bihar, India”

The manuscript reports on a well-executed experiment eliciting preferences for a small amount (20L) of delivered, high-quality water in Bihar, India. The paper is clear and well-written, the experiment is novel and, for the most part, the conclusions are policy-relevant and supported by the results. I found the work on experiential learning useful. I commend the authors for their transparency about methods and deviations from the registered research plan as well as the study’s limitations. These limitations are noted at the close of the article and include most importantly the fact that the sample is somewhat small and the samples were not randomized (and are unbalanced) across their two auctions. I have a number of comments and suggestions, though most are fairly minor. I denote comments that I would expect to see addressed in a revision (should the editor request one) with an asterisk. Those without an asterisk are collegial feedback and can be taken or left without prejudice.

• The authors seem primarily interested in the quality dimension of the service. I found myself interested in the delivery dimension, which would also save households collection time. This is largely unexplored here. The authors cite only one paper on this time dimension (Devoto et al). There are a large number of studies that have examined questions around water collection time, but the authors might want to at least cite two other high-quality, well-identified studies (Meeks et al 2017 and Gross et al 2018, refs provided at the end) and they may wish to at least reference the literature that examines revealed preference for time and quality by modeling households’ choice of source. Wagner et al. (2019) is a recent paper in this latter literature and includes a good review of the other existing work. *In the work at hand, if it was collected, it would be useful to at least report information on a) distance to the nearest shallow well (or do all households have wells on their compound?) and b) total water collection (to get a sense of how large a fraction a 20L delivery is)

• Line 159-164. I assume the 20L bottle and dispenser needed to participate in the program would of course have some alternate use as a normal storage container, should the household purchase the hardware but then discontinue with the delivery service. This isn’t discussed, and it would be helpful in this spot of the manuscript to perhaps report the going price for a “normal” 20L plastic jerrican that I would assume is ubiquitous in the region. This would help inform how much of a premium you are asking households to pay for the somewhat specialized part of the hardware.

• Section 2.3. I commend the authors for providing the complete BDM script for Auction 1 in the appendix.

o The included “tripwire” questions to test for understanding are great. As the authors may know, there have been some concerns raised recently about whether respondents understand BDM mechanisms. Buchardi et al (2022) in Uganda included some similar tripwire questions and found no problems with understanding. You might consider briefly reporting what you found – how many respondents got that first tripwire wrong and needed re-explanation - to contribute to this literature

o *I found Auction 2 a bit more confusing, so I suggest you also include the full experimental script for that auction as well.

o One part of Auction 2 that puzzled me was the inclusion of a new dimension of choice – the number of deliveries. You seem to have given respondents the opportunity to buy more than one delivery per day (unlike in Auction 1), but then this would require another purchase of a 20L tank and dispenser, correct? I suspect no one was interested in purchasing multiple deliveries, but you might consider reporting on this to help clarify.

o Line 301 “household-level covariates that may be endogenous to the bid price”. I found this language confusing. Endogenous makes me think of endogenous choices, but I think what you mean is that they are included to account for the imbalance in household characteristics between Auction 1 and Auction 2 because you didn’t randomize into those two treatments. Simpler language might be an improvement.

o Line 442-44 and Table 2.

Table 4 includes a covariate called “person can make decisions”. *Please report this in Table 2 and explain why the survey would have been conducted with someone who could not make decisions.

As noted above, a major concern for the study is external validity. Comparison here with some representative census data for Bihar would be very useful.

o Line 483: *I interpret your results also to mean that the 50% delivery discount had no effect on WTP: it was no different than WTP in Auction 1. This is apparent in the figures and the regression results, but unless I missed it, this is not highlighted in the text. Furthermore, several parts of the conclusions section (e.g. lines 619-620) seem to imply that your results show that modest subsidies can boost uptake. I may be wrong, but I don’t believe you found that. Instead, you found an experimental effect that when respondents in Auction 2 knew they might get a discount but then lost the lottery, it *lowered* WTP. By the way, I don’t think calling this a “negative externality” is quite correct, though I’m not sure of the correct term for the bad feeling you get when losing a lottery (experimental psychologist may have one).

o Stated preference: the design of the experiment seems sound and the status quo is sensibly constructed. There are, however, a few things I would consider non-standard that the authors might address or clarify.

*Most studies in this realm start from random utility theory, build out to an empirical model with assumptions about additive observable utility and iid type 1 errors for the unobserved components. They interpret model coefficients as utility differences. Most studies don’t spell all this out, but at least reference the RUM grounding. I believe the multinomial logistic approach used here is the same, but it might help some readers to make this connection. If the theoretical grounding is not RUM, please explain.

Although the delivery price was an attribute, the authors chose not to elicit WTP by including an upfront cost component (as in the Auctions). They return to this point in lines 670-672. Is there an explanation worth providing for why this wasn’t done?

Another non-standard feature is the construction of the choice sets in the experimental design. Researchers typically construct the full universe of choices and eliminate dominated alternatives, as you do here. But then we typically proceed to use software like Ngene to sample (and re-sample) from the remaining choices to construct a limited number of choice sets that have desirable properties like attribute balance and efficiency in identifying preference (i.e. D-efficiency). We then give all respondents that subset, or assign them in blocks to respondents. Here you simply randomly chose from the 124 choice sets in the field, which I’ve never seen done. This is probably defensible but comes at a cost of inefficiency and the possibility of attribute balance. *I would like to see at least a defense of this approach (perhaps referencing other work that does this) or an acknowledgement that it is non-standard so other researchers new to the literature don’t unthinkingly replicate it. I suspect it is fine, but I would also like to see an appendix table showing attribute balance in the choices actually shown to subjects. For example, among all the choice tasks actually chosen and completed, how often did Rs.0 appear? How often Rs. 6, 9, etc and how often safe to drink vs. not safe to drink?

*Is there a reason you don’t model delivery price as a continuous variable? I worry that perhaps it was not significant in some models as a continuous variable, namely Table 7 where the effect of Rs. 6 is larger than Rs. 9. Modeling as continuous would allow you to calculate part-worth WTP for convenience, safety and temperature.

A small point on Table 1: the layout confused me when I first looked at it. Rather than have four columns (which made me think there were four alternatives), you might just say “Price : 0,3,6 and 9 INR” and “Taste: Tastes nice(=1) or Tastes bad(=0).

*Please clarify what options respondents were given. At various points you say they could answer “Neither”, “no preference” or “prefer not to answer”. These are not equivalent and are three different preference statements. And I would simply report out and then drop the respondent who answered that he or she didn’t understand.

o Line 551-553. *I was confused by the language here. I would report separately the number of winners from Auction 1 and Auction 2. Also, it reads as if 66 people “won” the BDM auction but then only 56 actually got the service, so that 10 dropped out. This of course would violate the BDM procedures, since answers are binding. I think it is just wording, but please clarify.

o Line 574: on the percentage who chose the alternative vs. status quo: I don’t see where in the results this statement comes from. Models like these often have an “alternate specific constant” dummy to capture status quo choices, but your model doesn’t, nor do I see a constant.

o In the discussion of the perceptions of attributes: I think Figure 7 and Table 8 overlap enough and would suggest just presenting Table 8. I didn’t find this material particularly interesting. I’m also not sure (line 581) I would call these “stated preferences questions”. While it is literally true, that term tends to be reserved among folks in the field for a specific set of activities. These are just opinion questions.

o Line 602-3: Again I had trouble matching your statement here to specific results.

o Line 639-641. I agree with this statement, but there is a literature in the stated preference world on “test-retest” studies that is not cited here. A number of studies have re-surveyed the same households over time to see if preferences and WTP are stable. Brouwer et al (2016) is an example of a relatively recent one that has references to older studies.

o Lines 641-651, the discussion on purchasers vs. non-purchases is interesting and informative. These sentences mix together the effects (for noncustomers) of how the taste test and social marketing impacted preferences and the effects (for customers) of how the taste test, marketing AND product experienced shifted preferences. I was particularly interested in teasing out the effects of experience, so I’d consider re-writing this section to keep the effects clear in the readers’ mind.

References

Brouwer, R., I. Logar, and O. Sheremet. 2016. “Choice Consistency and Preference Stability in Test-Retests of Discrete Choice Experiment and Open-Ended Willingness to Pay Elicitation Formats.” Environmental and Resource Economics 68(3):1–23.

Burchardi, K.B., J. de Quidt, S. Gulesci, B. Lerva, and S. Tripodi. 2021. “Testing willingness to pay elicitation mechanisms in the field: Evidence from Uganda.” Journal of Development Economics 152(June):102701. Available at: https://doi.org/10.1016/j.jdeveco.2021.102701.

Gross, E., I. Guenther, Y. Schipper, and V. Der Walle. 2018. “Women are Walking and Waiting for Water: The Time Value of Public Water Supply.” Economic Development and Cultural Change.

Jalan, J., and E. Somanathan. 2008. “The importance of being informed: Experimental evidence on demand for environmental quality.” Journal of Development Economics 87:14–28.

Meeks, R. 2017. “Water works: The economic impact of water infrastructure.” Journal of Human Resources.

Wagner, J., J. Cook, and P. Kimuyu. 2019. “Household demand for water in rural Kenya.” Environmental and Resource Economics 74(4):1563–1584.

Reviewer #2: Summary: The authors work with an NGO to gather data regarding household preferences about a potable, clean water delivery service in Bihar, India. The authors use both revealed and stated preference methods to derive household willingness to pay for this delivery service and its associated hardware, and additionally evaluate 1) changes in household willingness to pay depending on getting or not getting a discounted price on delivery and 2) the importance and prioritization of different characteristics of water. They do so using an auction style revealed preference game and a stated preference discrete choice experiment. Noting the small sample size, the authors find that households that do not receive the discounted rate have a much lower willingness to pay for the service, and only a small portion of these households placed a positive bid. They also find that respondents’ stated preferences over taste of water and convenience of delivery change within one week of purchase.

Comments:

1. While both the WTP and DCE experiments are interesting, the WTP auction was very confusing to understand, and I would recommend making exact market prices, how much respondents know about market prices, and whether respondents are being given the market price clearer in the beginning of the paper. For example, the bottle and dispenser are 250 or 275 rupees in line 163. But Table 3 says the “market price of the good auctioned”, so that would be the hardware in Auction 2, is 285 rupees. In general, I think this section could use some wordsmithing to make the step-by-step process of the auction very clear.

2. Similarly, I think hypotheses 1 and 2 could be clearer. First, perhaps the authors could define “a” which I’ve understood is the difference in bids between Auction 1 and Auction 2 after removing the delivery fee in Auction 1, but is not immediately obvious. Second, it would be helpful to add a one line explanation of why we except a<35 in H1 and a>70 in H2.

3. A criteria in sampling selection was that the respondents were easily reachable by delivery drivers, meaning they are located along a main road. This means these are likely more affluent individuals (as the authors note in the discussion) and also that it is less likely that “negative learning” in terms of timely deliveries or warm water temperatures will occur. I wouldn’t generalize these results to a population in more rural settings.

4. In lines 491-494, I would recommend mentioning the lack of statistical significance in the groups that did receive the discount, with a short rationale for why that may occur.

5. Line 553 is a very confusing sentence I had to re-read multiple times.

6. PLOS authors have the option to publish the peer review history of their article (what does this mean?). If published, this will include your full peer review and any attached files.

Reviewer #1: No

Reviewer #2: No

---

## [Author Response · Author response to Decision Letter 0]

25 Oct 2022

Response to Reviewers

PONE-D-22-05048

 We thank the reviewers for their thorough review of our manuscript. In response to reviewer comments, we have made substantive changes throughout the manuscript, which are detailed in in-line responses to reviewer comments below. We are happy to address any further questions or inquiries. Data and underlying analysis files from this study will be made publicly available through the Harvard Dataverse upon acceptance. We have also uploaded our study protocol to AEA’s RCT Registry which provides its own DOI for the relevant study preregistration and protocol (https://doi.org/10.1257/rct.3829), rather than to protcols.io. If the editors prefer, we can also additionally upload this protocol document to protocols.io. 

 

Response to PLOS-ONE editorial comments

We have reviewed the style requirements and file naming guidelines and made appropriate adjustments. 

 Please include a complete copy of PLOS’ questionnaire on inclusivity in global research in your revised manuscript. Our policy for research in this area aims to improve transparency in the reporting of research performed outside of researchers’ own country or community. The policy applies to researchers who have travelled to a different country to conduct research, research with Indigenous populations or their lands, and research on cultural artefacts. The questionnaire can also be requested at the journal’s discretion for any other submissions, even if these conditions are not met. Please find more information on the policy and a link to download a blank copy of the questionnaire here: https://journals.plos.org/plosone/s/best-practices-in-research-reporting. Please upload a completed version of your questionnaire as Supporting Information when you resubmit your manuscript.

We have completed the PLOS questionnaire on inclusivity in global research and attached it to the revised manuscript as “S1 Checklist”. We have also included a new subsection 2.1.1. Inclusivity in global research on page 9 line 226-228: 

 “2.1.1. Inclusivity in global research

Additional information regarding the ethical, cultural, and scientific considerations specific to inclusivity in global research is included in the Supporting Information (S1 Checklist).”

This information has been corrected in the online submission.

4. Thank you for stating the following financial disclosure: "DBC; IND-19059; the International Growth Centre; https://www.theigc.org; The funders had no role in study design, data collection and analysis, decision to publish, or preparation of the manuscript.

DBC; Center for Global Public Health, University of California, Berkeley, School of Public Health; https://cgph.berkeley.edu; The funders had no role in study design, data collection and analysis, decision to publish, or preparation of the manuscript."

We note that one or more of the authors is affiliated with the funding organization, indicating the funder may have had some role in the design, data collection, analysis or preparation of your manuscript for publication; in other words, the funder played an indirect role through the participation of the co-authors. If the funding organization did not play a role in the study design, data collection and analysis, decision to publish, or preparation of the manuscript and only provided financial support in the form of authors' salaries and/or research materials, please do the following:

None of the authors is affiliated with any of the funding organizations in any way that would impact the analysis or manuscript preparation. We have revised the financial disclosure statement to read: 

“DBC; Center for Global Public Health (CGPH), University of California, Berkeley, School of Public Health; https://cgph.berkeley.edu. This is funding internal to the School of Public Health, with which DBC was affiliated at the time of the research; WHD is a faculty affiliate of the Center and does not receive salary contributions from the center. Other than CGPH providing travel support to DBC via a competitive internal grant process, the funders had no role in study design, data collection and analysis, decision to publish, or preparation of the manuscript. The specific roles of these authors are articulated in the ‘author contributions’ section."

a. Review your statements relating to the author contributions, and ensure you have specifically and accurately indicated the role(s) that these authors had in your study. These amendments should be made in the online form.

The form is changed as noted above.

b. Confirm in your cover letter that you agree with the following statement, and we will change the online submission form on your behalf:  “The funder provided support in the form of salaries for authors [insert relevant initials], but did not have any additional role in the study design, data collection and analysis, decision to publish, or preparation of the manuscript. The specific roles of these authors are articulated in the ‘author contributions’ section.

None of the authors of this manuscript drew a salary directly from any of the funding agencies. Granting agencies funded study activities, as well as providing travel and living expenses for author DBC only, but did not provide any salary. WHD, though affiliated with the Center for Global Public Health, does not receive salary reimbursement or other support from the center. 

See item 6 below. Once the manuscript is accepted for publication, the data files and finalized analysis files (.do files) will be made public on the Dataverse Repository and relevant accession numbers or DOIs to access the data will be provided.

Once the manuscript is accepted for publication, the data files and finalized analysis files (.do files) will be made public on the Dataverse Repository and relevant accession numbers or DOIs to access the data will be provided.

7. Please note that in order to use the direct billing option the corresponding author must be affiliated with the chosen institute. Please either amend your manuscript to change the affiliation or corresponding author, or email us at plosone@plos.org with a request to remove this option.

The corresponding Author, Drew Cameron, is affiliated with Yale University, which has an open access publishing agreement with PLoS journals. 

8. We note that you have referenced "Kremer M, Miguel EM, Mullainathan S, Null C, Zwane AP. Social Engineering: Evidence from a Suite of Take-up Experiments in Kenya. Unpublished Working Paper. 2011." which has currently not yet been accepted for publication. Please remove this from your References and amend this to state in the body of your manuscript: "Kremer M, Miguel EM, Mullainathan S, Null C, Zwane AP. Social Engineering: Evidence from a Suite of Take-up Experiments in Kenya. Unpublished Working Paper. 2011." as detailed online in our guide for authors http://journals.plos.org/plosone/s/submission-guidelines#loc-reference-style

We have removed this reference both from the body of the paper and the reference list for simplicity. 

9. We note you have included a table to which you do not refer in the text of your manuscript. Please ensure that you refer to Table 4 in your text; if accepted, production will need this reference to link the reader to the Table.

The in-text reference to Table 4 is on Page 26 line 563. It previously referred to Table 3 in error:

 “Table 4 and Fig 5 show the results of the two random price auctions.”

We found no retracted articles. We have made several changes to the reference list including the following: 

 Added the following 15 articles based on reviewer feedback:

 Brouwer R, Logar I, Sheremet O. Choice Consistency and Preference Stability in Test-Retests of Discrete Choice Experiment and Open-Ended Willingness to Pay Elicitation Formats. Environ Resour Econ. 2017;68(3):729-51.

 Chen J, Hui LS, Yu T, Feldman G, Zeng S, Ching TL, et al. Foregone Opportunities and Choosing Not to Act: Replications of Inaction Inertia Effect. Soc Psychol Personal Sci. 2021;12(3): 333-345. doi:10.1177/1948550619900570

 Gross E, Guenther I, Schipper Y. Women are Walking and Waiting for Water: The Time Value of Public Water Supply. Econ Dev Cult Change. 2018;66(3):489-517.

 International Institute for Population Sciences (IIPS) and ICF. 2021. National Family Health Survey (NFHS-5), India, 2019-21: Bihar. Mumbai: IIPS. URL: https://dhsprogram.com/pubs/pdf/FR374/FR374_Bihar.pdf

 International Institute for Population Sciences (IIPS) and ICF. 2021. National Family Health Survey (NFHS-5), India, 2019-21: District Fact Sheet, Supaul, Bihar. Mumbai: IIPS. URL: https://dhsprogram.com/pubs/pdf/OF43/BR_Supaul.pdf

 Liu, H, Chou H. The impact of different product formats on inaction inertia. J Soc Psychol. 2019;159(5): 546-560 DOI: 10.1080/00224545.2018.1520686

 Liu, T, Cheng T, Ni F. How consumers respond to the behavior of missing a free gift promotion: Inaction inertia effect on products offered as free gifts. J Soc Psychol. 2011;151(3): 361-81. 

 McFadden D. Conditional logit analysis of qualitative choice behavior. In: Zarembka P (ed). Frontiers in econometrics. Academic Press, New York. 1974:105-42.

 Meeks R. Water works: The economic impact of water infrastructure. J Hum Resour. 2017;52(4): 1119-1153

 Mørbak MR, Olsen SB. A within sample investigation of test-retest reliability in choice experiment surveys with real economic incentives. Aust J Agric Resour Econ. 2014;56:1-18.

 Tykocinski OE, Pittman TS. The consequences of doing nothing: Inaction inertia as avoidance of anticipated counterfactual regret. J Pers Soc Psychol. 1998;75(3): 607-16. DOI: 10.1037/0022-3514.75.3.607

 Tykocinski OE, Pittman TS, Tuttle EE. Inaction inertia: Forgoing future benefits as a result of an initial failure to act. J Pers Soc Psychol. 1995;68: 793-803. 

 Wagner J, Cook J, Kimuyu P. Household demand for water in rural Kenya. Environ Resour Econ. 2019;74(4):1563–1584.

 van Putten M, Zeelenberg M, van Dijk E, Tykocinski OE. Inaction inertia. Eur Rev Soc Psychol. 2013;24:1, 123-159, DOI: 10.1080/10463283.2013.841481

 Wright J, Dzodzomenyo M, Fink G, Wardrop NA, Aryeetey GC, Adanu RM, Hill AG. Subsidized Sachet Water to Reduce Diarrheal Disease in Young Children: A Feasibility Study in Accra, Ghana. Am J Trop Med Hyg. 2016;95(1):239-46. 

 Updated the following 3 citations to current versions from working papers or previous versions: 

 Adhvaryu AR. Learning, misallocation, and technology adoption: Evidence from new malaria therapy in Tanzania. Rev Econ Stud. 2014;81(4):1331-1365. doi: 10.1093/restud/rdu020.

 Berry J, Fischer G, Guiteras R. Eliciting and Utilizing Willingness- to-Pay: Evidence from Field Trials in Northern Ghana. J Polit Econ. 2020;128(4): 1436-73. 

 Jeuland M, McClatchey M, Patil SR, Pattanayak SK, Poulos CA, Yang JC. Do Decentralized Community Treatment Plants Provide Better Water? Evidence from Andhra Pradesh. Land Econ. University of Wisconsin Press. 2021;97(2): 345-371.

 Edited the following 27 citations to better match Vancouver Style

 Akerlof GA. The Market for “Lemons”: Quality Uncertainty and the Market Mechanism. Q J Econ. 1970;84(3):488-500.

 Ashraf N, Berry J, Shapiro J. Can higher prices stimulate product use? Evidence from a field experiment in Zambia. Am Econ Rev. 2010;100:2283-2413. 

 Becker GM, DeGroot MH, Marshak J. Measuring utility by a single-response sequential method. Behav Sci. 1964;9(3):226-32. 

 Bertrand M, Mullainathan S, Shafir E. Behavioral Economics and marketing in aid of decision making among the poor. J Public Policy Mark. 2006;25(1):8-23.

 Bridges JFP, Hauber AB, Marshall D, Lloyd A, Prosser LA, Regier DA, Johnson FR, Mauskopf J. Conjoint analysis applications in health—a checklist: A report of the ISPOR Good Research Practices for Conjoint Analysis Task Force. Value Health. 2011;14:403–13. 

 Burt Z, Njee RM, Mbatia Y, Msimbe V, Brown J, Clasen TF, Malebo HM, Ray I. User preferences and willingness to pay for safe drinking water: Experimental evidence from rural Tanzania. Soc Sci Med. 2017;173:63-71.

 Cameron D, Dow WH. The effect of short-term subsidies on demand for potable water in rural Bihar, India. AEA RCT Registry. 2021, Jan 19. Doi: 10.1257/rct.4323-6.0

 Cameron D, Dow WH. The effect of short-term subsidies on demand for potable water in rural Bihar, India: Pilot study. AEA RCT Registry. 2022, Oct 25. DOI:10.1257/rct.3829 

 Delaire C, Das A, Amrose S, Gadgil A, Roy J, Ray I. Determinants of the use of alternatives to arsenic-contaminated shallow groundwater: an exploratory study in rural West Bengal, India. J Water Health. 2017;15(5):799-812. 

 Devoto F, Duflo E, Dupas P, Pariente W, Pons V. Happiness on Tap: Piped Water Adoption in Urban Morocco. Am Econ J: Econ Policy. 2012;4(4):68-99. 

 Dupas P, Miguel E. Chapter 1 - Impacts and Determinants of Health Levels in Low-Income Countries. In: Banerjee AV, Duflo E, editors. Handbook of Economic Field Experiments. 2: North-Holland; 2017. p. 3-93.

 Fischer G, Karlan D, McConnell M, Raffler P. Short-term subsidies and seller type: A health products experiment in Uganda. J Dev Econ. 2019;137:110-24. 

 Murray CJL, Aravkin AY, Zheng P, Abbafati C, Abbas KM, Abbasi-Kangevari M, et al. Global burden of 87 risk factors in 204 countries and territories, 1990-2019: a systematic analysis for the Global Burden of Disease Study 2019. The Lancet. 2020;396(10258):1223-49.

 Hamoudi A, Jeuland M, Lombardo S, Patil S, Pattanayak SK, Rai S. The effect of water quality testing on household behavior: Evidence from an experiment in rural India. Am J Trop Med Hyg. 2012;87(1):18- 22. 

 Hauber AB, Gonzalez JM, Groothuis-Oudshoorn CG, Prior T, Marshall DA, Cunningham C, et al. Statistical Methods for the Analysis of Discrete Choice Experiments: A Report of the ISPOR Conjoint Analysis Good Research Practices Task Force. Value Health. 2016;19(4):300–15. 

 Johnson FR, Lancsar E, Marshall D, Kilambi V, Muhlbacher A, Regier DA, Bresnahn BW, Kanninen B, Bridges JFP. Constructing experimental designs for discrete-choice experiments: Report of the ISPOR conjoint analysis experimental design good research practices task force. Value Health. 2013;16(1):3–13. 

 Kremer M, Miguel EM. The illusion of sustainability. Q J Econ. 2007;122(3):1007-65. 

 Kremer M, Glennerster R. Chapter Four - Improving Health in Developing Countries: Evidence from Randomized Evaluations. In: Pauly MV, McGuire TG, Barros PP, editors. Handbook of Health Economics. 2: Elsevier; 2011. p. 201-315.

 Levine DI, Beltramo T., Blalock G, Cotterman C, Simons AM. What Impedes Efficient Adoption of Products? Evidence from Randomized Sales Offers for Fuel-Efficient Cookstoves. J Eur Econ Assoc. 2018;16(6):1850-80.

 Luoto J, Mahmud M, Albert J, Luby S, Najnin N, Unicomb L, Levine D. Learning to dislike safe water products: results from a randomized controlled trial of the effects of direct and peer experience on willingness to pay. Environ Sci Technol. 2012;46(11):6244e6251. 

 Munshi K, Myaux J. Social norms and the fertility transition. J Dev Econ. 2005;80:1-38. 

 Nelson P. Information and Consumer Behavior. J Polit Econ. 1970;78(2):311-29. 

 Orgill J, Shaheed A, Brown J, Jeuland M, Water quality perceptions and willingness to pay for clean water in peri-urban Cambodian communities. J Water Health. 2013;11(3):489. 

 Oster E, Thornton R. Determinants of technology adoption: Peer effects in menstrual cup take-up. J Eur Econ Assoc. 2012;10(6):1263-93. 

 Poulos C, Yang JC, Patil SR, Pattanayak S, Wood S, Goodyear L, Gonzalez JM. Consumer preferences for household water treatment products in Andhra Pradesh, India. Soc Sci Med. 2012;75(4):738-46.

 Ray I, Smith K. Towards safe drinking water and clean cooking for all. Lancet Glob Health. 2021;9:e361. 

 Ryan M, Farrar S. Using conjoint analysis to elicit preferences for health care. Br Med J. 2000;320(7248):1530-3. 

 Singh SK, Ghosh AK, Kumar A, Kislay K, Kumar C, Tiwari RR, Parwez R, Kumar N, Imam MD. Groundwater Arsenic Contamination and Associated Health Risks in Bihar, India. Int J Environ Res. 2014;8(1):49-60. 

 Removed the following 2 unnecessary citations 

 Kremer M, Miguel EM, Mullainathan S, Null C, Zwane AP. Social Engineering: Evidence from a Suite of Take-up Experiments in Kenya. Unpublished Working Paper. 2011. 

 Zanolini A, Sikombe K, Sikazwe I, Eshun-Wilson I, Somwe P, Moore CB, Topp SM, Czaicki N, Beres LK, Mwamba CP, Padian N, Holmes CB, Geng EH. Understanding preferences for HIV care and treatment in Zambia: Evidence from a discrete choice experiment among patients who have been lost to follow-up. PLOSMed. 2018;15(8):e1002636. 

 

Reviewer 1 Summary and Comments

Reviewer #1: Review of “Product preferences and willingness to pay for potable water delivery: Experimental evidence from rural Bihar, India”  

The manuscript reports on a well-executed experiment eliciting preferences for a small amount (20L) of delivered, high-quality water in Bihar, India. The paper is clear and well-written, the experiment is novel and, for the most part, the conclusions are policy-relevant and supported by the results. I found the work on experiential learning useful. I commend the authors for their transparency about methods and deviations from the registered research plan as well as the study’s limitations. These limitations are noted at the close of the article and include most importantly the fact that the sample is somewhat small and the samples were not randomized (and are unbalanced) across their two auctions. I have a number of comments and suggestions, though most are fairly minor. I denote comments that I would expect to see addressed in a revision (should the editor request one) with an asterisk. Those without an asterisk are collegial feedback and can be taken or left without prejudice.  

We would like to thank the reviewer for their thorough and very generous feedback of our manuscript. Your comments are extremely helpful and will certainly improve our final product. We would be very happy to include you in our acknowledgements but understand if you prefer to remain anonymous. 

• The authors seem primarily interested in the quality dimension of the service. I found myself interested in the delivery dimension, which would also save households collection time. This is largely unexplored here. The authors cite only one paper on this time dimension (Devoto et al). There are a large number of studies that have examined questions around water collection time, but the authors might want to at least cite two other high-quality, well-identified studies (Meeks et al 2017 and Gross et al 2018, refs provided at the end) and they may wish to at least reference the literature that examines revealed preference for time and quality by modeling households’ choice of source. Wagner et al. (2019) is a recent paper in this latter literature and includes a good review of the other existing work. *In the work at hand, if it was collected, it would be useful to at least report information on a) distance to the nearest shallow well (or do all households have wells on their compound?) and b) total water collection (to get a sense of how large a fraction a 20L delivery is) 

We have added clarifying statements about the current burden of household water collection. We have added particular reference to the sources you provided on page(s) 3-4 lines 88-90: 

“Although many studies already examine willingness to pay (WTP) for water treatment products (Null et al. 2012; Ahuja et al. 2010), as well as the concomitant burden of water collection time (Meeks et al. 2017; Gross et al. 2018) as well as quality (Wagner et al. 2019), very little evidence exists for fully treated delivery in low-income rural communities.”

With regard to the time spent in water collection (burden), we have added language about the status quo in Bihar generally on page 4 (lines 95-101): 

“As of 2020, 85.4% of rural households in the study region use ubiquitous (mostly government-built) shallow well hand pumps as their primary drinking water source (IIPS 2021), a slight improvement over the 93% who primarily used this source less than a decade earlier (Das et al 2013). Nearly all (99%) rural households have access to these or better sources within their household plots, meaning that time spent collecting water is minimal, and most (94.2%) report using no water treatment (Das et al. 2013; IIPS 2021).”

as well as the status quo burden of water collection within our sample on page 11 (lines 246-254):

“Despite being poorer on average, water use patterns among study participants are well in-line with normal practices for rural Bihar. All study households reported primarily using hand pumps with shallow wells as their principal source of water for drinking, cooking, and bathing. As with the rest of Bihar, the status quo time burden of water collection in our sample is minimal with all but 2 households having primary water sources in their homes or yards. In total, 17.9% of households self-reported ever treating their water before use – almost all of whom used boiling as the primary treatment strategy (IIPS 2021a; IIPS 2021b). Since water is collected a la carte by almost all households, we do not have data on the daily total volume of water collected or consumed by households.”

as well as with the addition of the new Table 1 on page 10. 

• Line 159-164. I assume the 20L bottle and dispenser needed to participate in the program would of course have some alternate use as a normal storage container, should the household purchase the hardware but then discontinue with the delivery service. This isn’t discussed, and it would be helpful in this spot of the manuscript to perhaps report the going price for a “normal” 20L plastic jerrican that I would assume is ubiquitous in the region. This would help inform how much of a premium you are asking households to pay for the somewhat specialized part of the hardware. 

Thank you. On page 7 lines 185-188 we now elaborate on the current use of bottles and alternative usages: 

“These bottles and dispensers are ubiquitous in the area and used as the primary water storage device for those purchasing treated water from any local vendor. Their normal retail price is widely known. Notably, we also found that these storage bottles were sometimes repurposed by customers who discontinued service to store grains and other dry foods to prevent spoiling from mold.”

We provide added clarity about prices in lines 182-184 of the same page: 

“By the time of this study, all SHRI customers of the water service had to purchase a bottle and dispenser (the hardware) either from SHRI at a wholesale price of ₹250, or from another vendor in the market where a single bottle and dispenser normally retails for ₹275.”

• Section 2.3. I commend the authors for providing the complete BDM script for Auction 1 in the appendix. 

o The included “tripwire” questions to test for understanding are great. As the authors may know, there have been some concerns raised recently about whether respondents understand BDM mechanisms. Buchardi et al (2022) in Uganda included some similar tripwire questions and found no problems with understanding. You might consider briefly reporting what you found – how many respondents got that first tripwire wrong and needed re-explanation - to contribute to this literature 

We now provide greater detail on pages 27-8, lines 576-83: 

“Additional results for both auctions can again be found in Table 4, including exploring whether initial bids matched final bids along with the amounts of bid difference, as well as any feelings of regret expressed by those who lost the auction. Results show between 61%-82% agreement in bids between rounds, with low levels of expressed regret among auction losers. Anecdotally, study respondents were highly engaged and excited to play the auction “game.””

This passage refers to new data added to Table 4, page 26, final 6 rows of table. Unfortunately, tripwire questions about the bar of soap were recorded inconsistently by enumerators, and these data are not reliable. 

o *I found Auction 2 a bit more confusing, so I suggest you also include the full experimental script for that auction as well. 

Thank you for noticing this omission! We have included this second auction script, which is now in the Appendix on pages 54-57. 

o One part of Auction 2 that puzzled me was the inclusion of a new dimension of choice – the number of deliveries. You seem to have given respondents the opportunity to buy more than one delivery per day (unlike in Auction 1), but then this would require another purchase of a 20L tank and dispenser, correct? I suspect no one was interested in purchasing multiple deliveries, but you might consider reporting on this to help clarify. 

This was a mistake in writing – thank you for catching! This was not an explicit difference between the two auctions. We have moved and slightly edited the line starting on page 12 line 308, to now reside on page 13 lines 319-322:

“In either auction, households could have purchased more than one delivery per day or could elect to have seven deliveries spread over more than seven days. In either case a maximum of seven filled bottles were allowed among winners. No households elected to receive more than one delivery per day.”

o Line 301 “household-level covariates that may be endogenous to the bid price”. I found this language confusing. Endogenous makes me think of endogenous choices, but I think what you mean is that they are included to account for the imbalance in household characteristics between Auction 1 and Auction 2 because you didn’t randomize into those two treatments. Simpler language might be an improvement. 

 Here we have removed the phrase “that may be endogenous to the bid price” for clarity. See page 16 line 387. The line now reads: 

"X_hn is a vector of household-level covariates, η_n is a vector of neighborhood-fixed effects, and ϵ_hs is an error term clustered at the structure-level.”

o Line 442-44 and Table 2. 

Table 4 includes a covariate called “person can make decisions”. *Please report this in Table 2 and explain why the survey would have been conducted with someone who could not make decisions. 

This language was an error in transcription from the codebook. Our selection criteria are now explained on page 8 lines 202-4: 

“Participants could not be current SHRI water users, had to be allowed to make purchase decisions for the households, and had to be willing to participate in study activities.”

The variable previously in Table 4 (now Table 5) now reads: 

 “Respondent typically makes purchase decisions”

This variable is now also included in the balance Table 3, on page 24. 

As noted above, a major concern for the study is external validity. Comparison here with some representative census data for Bihar would be very useful. 

We have provided this in an updated Table 1 (page 10). We have also included the following passage on page 9 lines 231-41:

“Reported median monthly income was ₹9000 ($121.62). Most households, 77.8%, were Hindu while 22.2% were Muslim. Despite being connected to the road system, compared to the general population of rural Bihar, our sample was slightly less socioeconomically advantaged. Table 1 compares several key demographic and household characteristics between households in this study and those in wider rural Bihar. Most notably, several indicators are worse than average for rural Bihar, including the level of education of household heads (compared to the same statistic for all men and women in Bihar), the percent of households practicing open defecation, percent of household structures made of improved materials (floors, walls and roofs), and the percent of children under 5 having experienced diarrhea in the past two weeks (IIPS 2021). Because this sample is slightly poorer than average, our results may have a slight downward bias with regard to product adoption.”

o Line 483: *I interpret your results also to mean that the 50% delivery discount had no effect on WTP: it was no different than WTP in Auction 1. This is apparent in the figures and the regression results, but unless I missed it, this is not highlighted in the text. Furthermore, several parts of the conclusions section (e.g. lines 619-620) seem to imply that your results show that modest subsidies can boost uptake. I may be wrong, but I don’t believe you found that. Instead, you found an experimental effect that when respondents in Auction 2 knew they might get a discount but then lost the lottery, it *lowered* WTP. By the way, I don’t think calling this a “negative externality” is quite correct, though I’m not sure of the correct term for the bad feeling you get when losing a lottery (experimental psychologist may have one).  

Thank you for your comment here, which is central to our findings. We have made the following changes throughout the manuscript to reflect this interpretation. The abstract (lines 50-6) now reads:

“We also find mixed evidence on the effect of small price subsidies for various parts of the delivery service, and that one week of initial participation leads to significant changes in stated preferences for the taste of the treated water as well as the convenience of the delivery service. While more evidence is needed on the effect of subsidies, our findings suggest that marketing on taste and convenience could help increase uptake of clean water delivery services in rural and last-mile communities that have yet to receive piped water.”

In the findings section, page 28 lines 656-8 now read: 

“In both models, we find that receiving a discount had no effect on the overall bid price among winners, suggesting that an unexpected, small promotional discount was not effective at increasing demand for the overall service.”

And page 38 lines 820-4 now read: 

“However, our subsequent analysis suggests that this initial difference is misleading, and likely driven by a decrease in mean WTP for hardware among those who missed out on discounts in Auction 2. Our findings led to a failure to reject alternative hypothesis 2, suggesting that the effect of missing out on small discounts had modest but larger negative effects on demand.”

In the discussion section on page 39 lines 841-44 

“First, although we find that there is latent demand for fully treated and delivered water, this demand may not be easily increased using very small promotional subsidies for the service alone.”

and page 39 lines 850-61, with regard to the overall effects and to the previously termed “negative externalities” (line 857) now read: 

“Though we cannot generalize beyond this small-n study, our results can be seen as hypothesis-generating for designing interventions for delivered water services. For example, mean WTP for hardware among those who received a 50% water delivery discount (equivalent to $0.49 USD) for a short period (roughly one week) suggests that subsidies for water delivery might have scope to increase initial WTP for the requisite hardware, relative only to those who do not receive a discount. Indeed, these gains may not be driven by overall demand for the service, but instead by a dampening effect on demand among those who missed out on the small discount – akin to a phenomenon that social psychologists have termed the inaction inertia effect (Tykocinski et al. 1995; Tykocinski and Pittman 1998; Liu et al. 2011; van Putten et al. 2013; Liu and Chou 2019; Chen et al. 2021). Increasing the size or lengthening the time frame of this discount could lead to a higher overall share of individuals purchasing water delivery, and positively learning about product qualities during a trial period compared to the status quo.”

With the addition of several new references related to the term ‘inaction inertia effect’ in the bibliography:

Chen J, Hui LS, Yu T, et al. (2021). “Foregone Opportunities and Choosing Not to Act: Replications of Inaction Inertia Effect.” Social Psychological and Personality Science, 12(3): 333-345. doi:10.1177/1948550619900570

Liu, Hsin-Hsien & Hsuan-Yi Chou. (2019). “The impact of different product formats on inaction inertia,” Journal of Social Psychology, 159(5): 546-560 DOI: 10.1080/00224545.2018.1520686

Liu, Tsung-Chi, Ti Cheng and Feng-Yu Ni. (2011). “How consumers respond to the behavior of missing a free gift promotion: Inaction inertia effect on products offered as free gifts.” Journal of Social Psychology, 151(3): 361-81. 

Marijke van Putten, Marcel Zeelenberg, Eric van Dijk & Orit E. Tykocinski. (2013). “Inaction inertia.” European Review of Social Psychology, 24:1, 123-159, DOI: 10.1080/10463283.2013.841481

Tykocinski, Orit E and TS Pittman. (1998). “The consequences of doing nothing: Inaction inertia as avoidance of anticipated counterfactual regret.” Journal of Personality and Social Psychology, 75(3): 607-16. DOI: 10.1037/0022-3514.75.3.607

Tykocinski, OE, TS Pittman and EE Tuttle. (1995). “Inaction inertia: Forgoing future benefits as a result of an initial failure to act.” Journal of Personality and Social Psychology, 68: 793-803. 

o Stated preference: the design of the experiment seems sound and the status quo is sensibly constructed. There are, however, a few things I would consider non-standard that the authors might address or clarify. 

*Most studies in this realm start from random utility theory, build out to an empirical model with assumptions about additive observable utility and iid type 1 errors for the unobserved components. They interpret model coefficients as utility differences. Most studies don’t spell all this out, but at least reference the RUM grounding. I believe the multinomial logistic approach used here is the same, but it might help some readers to make this connection. If the theoretical grounding is not RUM, please explain. 

Thank you for noting. The passage on page 20 lines 476-8 now reads:

“To answer these questions preferences for policy alternatives selected in our discrete choice experiment are modeled based on the Random Utility Model (McFadden 1974). Econometrically, we analyze…”

Although the delivery price was an attribute, the authors chose not to elicit WTP by including an upfront cost component (as in the Auctions). They return to this point in lines 670-672. Is there an explanation worth providing for why this wasn’t done? 

We agree that this would have been valuable, but we did not pursue this due to the already complicated field design. We mention this more explicitly now on pages 42-3 lines 885-8” 

“Through our DCE we do find that preferences for price remain relatively stable in both groups over time, but due to the difficulty of modifying the DCE mid-study, these findings are only specific to the price of water alone, as pre-specified.”

Another non-standard feature is the construction of the choice sets in the experimental design. Researchers typically construct the full universe of choices and eliminate dominated alternatives, as you do here. But then we typically proceed to use software like Ngene to sample (and re-sample) from the remaining choices to construct a limited number of choice sets that have desirable properties like attribute balance and efficiency in identifying preference (i.e. D-efficiency). We then give all respondents that subset, or assign them in blocks to respondents. Here you simply randomly chose from the 124 choice sets in the field, which I’ve never seen done. This is probably defensible but comes at a cost of inefficiency and the possibility of attribute balance. *I would like to see at least a defense of this approach (perhaps referencing other work that does this) or an acknowledgement that it is non-standard so other researchers new to the literature don’t unthinkingly replicate it. I suspect it is fine, but I would also like to see an appendix table showing attribute balance in the choices actually shown to subjects. For example, among all the choice tasks actually chosen and completed, how often did Rs.0 appear? How often Rs. 6, 9, etc and how often safe to drink vs. not safe to drink? 

Our approach was definitely non-standard. We modify our language on page 30 lines 673-6 which now read: 

“Our approach was non-standard and did not use DCE software to achieve attribute balance and D-efficiency. Nonetheless, attribute levels were well balanced in each of the two surveys, as can be found in Appendix Table 1.”

We have also included a new Appendix Table 1 on page 59 showing balance in choices presented, as referenced above. 

*Is there a reason you don’t model delivery price as a continuous variable? I worry that perhaps it was not significant in some models as a continuous variable, namely Table 7 where the effect of Rs. 6 is larger than Rs. 9. Modeling as continuous would allow you to calculate part-worth WTP for convenience, safety and temperature. 

We pre-specified modeling price as a categorical variable in our analysis to match the choice set provided to respondents and to keep models simple to interpret, so there was no p-hacking here. We have not tried modeling this as a continuous variable.

A small point on Table 1: the layout confused me when I first looked at it. Rather than have four columns (which made me think there were four alternatives), you might just say “Price : 0,3,6 and 9 INR” and “Taste: Tastes nice(=1) or Tastes bad(=0). 

We have reorganized this table to have only one column of levels corresponding to each attribute. We now include the language used to describe each attribute to respondents as well as a note in the table for greater clarity. See new Table 2 on page 19. 

*Please clarify what options respondents were given. At various points you say they could answer “Neither”, “no preference” or “prefer not to answer”. These are not equivalent and are three different preference statements. And I would simply report out and then drop the respondent who answered that he or she didn’t understand.  

Thank you for this note – there were errors in transcription in the manuscript which have been addressed. To avoid confusion, page 30 lines 678-686 now reads: 

“At baseline, responses were as follows: ‘alternative’ (308), ‘status quo’ (150), ‘no preference’ (27) and ‘do not understand’ (1) – this respondent was dropped. A total of 59 households (36.4%) chose the alternative scenario in all three draws, 21 households (13%) always chose the status quo, one household chose ‘no preference’ all three times, and the remaining 81 households (50%) varied their responses. At follow-up, a total of three possible alternative scenarios were not generated. The total number of possible endline responses were as follows: ‘alternative’ (299), ‘status quo’ (176), ‘no preference’ (5), ‘do not understand’ (0). A total of 70 households (43.8%) always chose the alternative, 34 (21.3%) always chose the status quo, and the remaining 56 (35.0%) varied their responses.”

o Line 551-553. *I was confused by the language here. I would report separately the number of winners from Auction 1 and Auction 2. Also, it reads as if 66 people “won” the BDM auction but then only 56 actually got the service, so that 10 dropped out. This of course would violate the BDM procedures, since answers are binding. I think it is just wording, but please clarify. 

We have modified the language reporting on the organization of Figure 5 and the results of of the auction on pages 26-7 lines 570-628: 

“The panels on the left (I and IV) show all the bids placed by those who played the auction, while the right panels (II and III) show the bids of only those who “won” the respective auctions. 

“In Auction 1, respondents were asked to bid on a package of seven water deliveries plus requisite hardware. The average bid was ₹164 among all bidders (Panel I) and ₹222 among all non-zero bidders (Panel II). In Auction 2 (panels III and IV), respondents were first asked to draw a number from a hat with a 50:50 chance to receive a 50% discount on the price of 7 water deliveries. The dashed lines show the WTP among those who received the ₹35 discounts, while the dotted lines show the WTP among those who did not. In Panel IV, the average bid among winners was ₹138, and among non-winners was ₹55. In Panel III, the average (non-zero) bid among winners was ₹220 and the average non-zero bid among non-winners was ₹186. Additional results for both auctions can again be found in Table 4, including exploring whether initial bids matched final bids along with the amounts of bid difference, as well as any feelings of regret expressed by those who lost the auction. Results show between 61%-82% agreement in bids between rounds, with low levels of expressed regret among auction losers. Anecdotally, study respondents were highly engaged and excited to play the auction “game.””

Regarding the 10 who dropped, we have also modified the language in this paragraph (now) starting on Page 27 lines 629-638 for greater clarity, which now reads: 

“Because the WTP auctions represent a lab-in-the-field experiment, there were some violations to the rules of the game for which we could not control. Out of the 66 winners from either auction, a total of 10 respondents – 6 in Auction 1 and 4 in Auction 2 – later refused to pay for the hardware and delivery service and recanted on the rules of the game before deliveries could commence (all those who recanted in Auction 2 had randomly drawn a 50% discount on water deliveries). Because this was a vulnerable population, we made exceptions for these families to recoup the money they had agreed to spend. But we did not advertise this practice. The results we show in Tables 4 and 5 and in Figures 5 and 6 include the original bids from these 10 respondents who later recanted. When we later examine customer preferences starting in section 3.2, these 10 winners are counted among “non-customers.””

In short, we went to great lengths to figure out whether these 10 households might have had an alternate understanding of the rules or may have been trying to game the system (NB. Despite the rules of the game, this was a vulnerable population, and we could not ethically “bind” respondents to paying a price it turns out they could not afford). We are unable to assign any motive to their decision to recant, thus we include the winning bids of these 10 households in our analysis of the BDM auction as we have no reason to think that the prices, they bid did not reflect the value that they placed on the products. 

o Line 574: on the percentage who chose the alternative vs. status quo: I don’t see where in the results this statement comes from. Models like these often have an “alternate specific constant” dummy to capture status quo choices, but your model doesn’t, nor do I see a constant. 

The model does indeed include alternative-specific constants, though they are suppressed from the tables (a table note has now been added to indicate this). See DCE Tables 6, 7 and 8 (as well as in the appendix) on pages 32, 35 and 36 (Appendix pages 56, 60 and 61). 

o In the discussion of the perceptions of attributes: I think Figure 7 and Table 8 overlap enough and would suggest just presenting Table 8. I didn’t find this material particularly interesting. I’m also not sure (line 581) I would call these “stated preferences questions”. While it is literally true, that term tends to be reserved among folks in the field for a specific set of activities. These are just opinion questions. 

We agree that the findings are redundant from this figure. We have removed figure 7. We moved our previous summary of these preference questions to page 34 lines 776 to 783 before discussing the results from Table 9. (previously T8) on page 34, lines 783 to 785. 

Regarding the references to stated preferences questions, we agree with the terminology being inappropriate and have changed the passage on page 22 line 522:

“We also examine the mean difference in responses to opinion questions about the product between these two groups for another set of endline questions about product characteristics.”

and page 34, lines 783-5: 

“Table 9 shows the results of this set of opinion questions compared between customers and non-customers.”

o Line 602-3: Again I had trouble matching your statement here to specific results. 

We have revised this portion of the manuscript to reflect the results discussed earlier. See page 38 lines 820-4: 

“However, our subsequent analysis suggests that this initial difference is misleading, and likely driven by a decrease in mean WTP for hardware among those who missed out on discounts in Auction 2. Our findings led to a failure to reject alternative hypothesis 2, suggesting that the effect of missing out on small discounts had modest but larger negative effects on demand.”

o Line 639-641. I agree with this statement, but there is a literature in the stated preference world on “test-retest” studies that is not cited here. A number of studies have re-surveyed the same households over time to see if preferences and WTP are stable. Brouwer et al (2016) is an example of a relatively recent one that has references to older studies. 

Thanks for these, we have incorporated these references into the passage. See page 40 lines 875-79: 

“Although an experimental test-retest literature that examine the stability of preferences and WTP using discrete choice methods is well-established (Mørbak and Olsen 2014; Brouwer et al. 2017), to our knowledge, our stated preferences exercise is among the first examples of a DCE with the same sample conducted before and after the introduction of a water product in an LMIC setting.”

o Lines 641-651, the discussion on purchasers vs. non-purchases is interesting and informative. These sentences mix together the effects (for noncustomers) of how the taste test and social marketing impacted preferences and the effects (for customers) of how the taste test, marketing AND product experienced shifted preferences. I was particularly interested in teasing out the effects of experience, so I’d consider re-writing this section to keep the effects clear in the readers’ mind.  

We agree that this passage was confusing. We have re-written now on 882-7, page 40):

“According to self-reported opinion questions, many more respondents reported liking the taste of the delivered water than they did the safety of the product at endline (among especially customers, but also to a lesser extent among non-customers). By contrast, the results of the DCE suggest that safety and taste were roughly equally weighted as important drivers of demand among non-customers at endline. Meanwhile, like the self-reported opinion results, customers expressed a much greater preference for taste than safety at endline.”

References 

 Brouwer, R., I. Logar, and O. Sheremet. 2016. “Choice Consistency and Preference Stability in Test-Retests of Discrete Choice Experiment and Open-Ended Willingness to Pay Elicitation Formats.” Environmental and Resource Economics 68(3):1–23. 

 Burchardi, K.B., J. de Quidt, S. Gulesci, B. Lerva, and S. Tripodi. 2021. “Testing willingness to pay elicitation mechanisms in the field: Evidence from Uganda.” Journal of Development Economics 152(June):102701. Available at: https://doi.org/10.1016/j.jdeveco.2021.102701. 

 Gross, E., I. Guenther, Y. Schipper, and V. Der Walle. 2018. “Women are Walking and Waiting for Water: The Time Value of Public Water Supply.” Economic Development and Cultural Change. 

 Jalan, J., and E. Somanathan. 2008. “The importance of being informed: Experimental evidence on demand for environmental quality.” Journal of Development Economics 87:14–28. 

 Meeks, R. 2017. “Water works: The economic impact of water infrastructure.” Journal of Human Resources. 

 Wagner, J., J. Cook, and P. Kimuyu. 2019. “Household demand for water in rural Kenya.” Environmental and Resource Economics 74(4):1563–1584.

These have been incorporated into the bibliography and referenced in the body of the manuscript.

 

Reviewer 2 Comments

Reviewer #2: Summary: The authors work with an NGO to gather data regarding household preferences about a potable, clean water delivery service in Bihar, India. The authors use both revealed and stated preference methods to derive household willingness to pay for this delivery service and its associated hardware, and additionally evaluate 1) changes in household willingness to pay depending on getting or not getting a discounted price on delivery and 2) the importance and prioritization of different characteristics of water. They do so using an auction style revealed preference game and a stated preference discrete choice experiment. Noting the small sample size, the authors find that households that do not receive the discounted rate have a much lower willingness to pay for the service, and only a small portion of these households placed a positive bid. They also find that respondents’ stated preferences over taste of water and convenience of delivery change within one week of purchase.  

Comments: 

 While both the WTP and DCE experiments are interesting, the WTP auction was very confusing to understand, and I would recommend making exact market prices, how much respondents know about market prices, and whether respondents are being given the market price clearer in the beginning of the paper. For example, the bottle and dispenser are 250 or 275 rupees in line 163. But Table 3 says the “market price of the good auctioned”, so that would be the hardware in Auction 2, is 285 rupees. In general, I think this section could use some wordsmithing to make the step-by-step process of the auction very clear.  

Thank you for this comment. We agree that the auction and market conditions were confusing. We have done the following to address these concerns: 

- We’ve changed the paragraph on page 7 lines 176-88 to read: 

“The per unit price sometimes rises as high as ₹15 ($0.21) during summer months when the plant uses a chiller to cool the water prior to delivery. Reverse osmosis water produced and delivered by other vendors usually retails for between ₹15 and ₹20 in the same neighborhoods. Before 2019, SHRI customers paid an additional (refundable) ₹120 deposit for each bottle and dispenser (see Fig 1). However, because of high rates of theft, damage and loss of these bottles and dispensers, the hardware was sold to participating households starting in early 2019. By the time of this study, all SHRI customers of the water service had to purchase a bottle and dispenser (the hardware) either from SHRI at a wholesale price of ₹250, or from another vendor in the market where a single bottle and dispenser normally retails for ₹275. These bottles and dispensers are ubiquitous in the area and used as the primary water storage device for those purchasing treated water from any local vendor. Their normal retail price is widely known. Notably, we also found that these storage bottles were sometimes repurposed by customers who discontinued service to store grains and other dry foods to prevent spoiling from mold.”

- We have changed the reference to rs. 285 in Table 3 (now table 4) to fix the error. It now reads rs. 250 and has an additional line (row 2) enumerating the “additional price faced for deliveries” for either of the “discount” or “no discount” scenarios in Auction 2. See Table 4 on page 25. 

2. Similarly, I think hypotheses 1 and 2 could be clearer. First, perhaps the authors could define “a” which I’ve understood is the difference in bids between Auction 1 and Auction 2 after removing the delivery fee in Auction 1, but is not immediately obvious. Second, it would be helpful to add a one line explanation of why we except a<35 in H1 and a>70 in H2.  

Thank you for these suggestions – we agree the hypotheses could be clearer. We have made the following changes:

- There was an error in transcription here referring to a=70. The passage on page 15 line 368 for H0 now reads:

“… b=70 in the no discount group.”

- For greater clarity we have defined our a and b terms and amended the passage on page 14 lines 352-59 to read: 

“To answer this, we test the following hypotheses as represented in Fig 3. In this figure, x represents the willingness to pay for the whole delivery service as a package (bottle + dispenser + deliveries of water). The value a represents the change in demand placed on the overall package that results when consumers are met with an unexpected discount of ₹35 on the price of water deliveries – or the effect on demand of winning. Meanwhile, the value b represents the corresponding change in demand for the overall package that results when consumers knowingly do not win a discount of ₹35 on the price of water deliveries – or, the effect on demand of missing out.”

- We have also amended the hypothesis statements on page 15 to provide greater clarity:

H1. Lines 372-3: “We would expect this if the effect of winning has a disproportionately large positive impact on demand for the delivery package.”

H2. Lines 377-9: “We would expect this if the effect of missing out has a disproportionately large negative impact on demand for the delivery package.”

3. A criteria in sampling selection was that the respondents were easily reachable by delivery drivers, meaning they are located along a main road. This means these are likely more affluent individuals (as the authors note in the discussion) and also that it is less likely that “negative learning” in terms of timely deliveries or warm water temperatures will occur. I wouldn’t generalize these results to a population in more rural settings.  

We have addressed this by changing the passage on page 43 lines 946-7 which now reads: 

“Our respondents lived close to the main road, which is often a sign of higher economic status than the community at large (Chambers 1980); thus, results have limited generalizability to those who are less socioeconomically advantaged as well as those not living directly on a main road with easy access to delivery.”

4. In lines 491-494, I would recommend mentioning the lack of statistical significance in the groups that did receive the discount, with a short rationale for why that may occur.  

We have appended this paragraph to include the following sentence, see page 28, lines 656-8: 

“In both models, we find that receiving a discount had no effect on the overall bid price among winners, suggesting that an unexpected, small promotional discount was not effective at increasing demand for the overall service.” 

5. Line 553 is a very confusing sentence I had to re-read multiple times.

We have re-written this passage for clarity now on page 33 Lines 738-742. It now reads: 

“This group of 56 eventual customers has the highest willingness to pay for the product and experienced the product through purchase of the hardware and 7 deliveries. We examine the difference in stated preferences between these customers (who did experience the product through purchasing water deliveries) and non-customers (who did not) below.”

---

## [Decision Letter · Decision Letter 1]

16 Jan 2023

PONE-D-22-05048R1Product preferences and willingness to pay for potable water delivery: Experimental evidence from rural Bihar, IndiaPLOS ONE

Dear Dr. Cameron,

Thank you for submitting the revised version of your manuscript. After careful consideration, we feel that it has merit and you have done a good job responding to the reviewers' comments. While the first reviewer has no further comments, reviewer #2 has some minor ones. I believe you should be able to address these minor points in no time. I very much look forward to reading the revised version of your paper, after which I hope to make the final decision on your submission.

We look forward to receiving your revised manuscript.

Kind regards,

Ilke Onur, Ph.D.

Academic Editor

PLOS ONE

Journal Requirements:

Reviewers' comments:

Reviewer's Responses to Questions

**Comments to the Author**

1. If the authors have adequately addressed your comments raised in a previous round of review and you feel that this manuscript is now acceptable for publication, you may indicate that here to bypass the “Comments to the Author” section, enter your conflict of interest statement in the “Confidential to Editor” section, and submit your "Accept" recommendation.

Reviewer #1: All comments have been addressed

Reviewer #2: (No Response)

2. Is the manuscript technically sound, and do the data support the conclusions?

Reviewer #1: (No Response)

Reviewer #2: Yes

3. Has the statistical analysis been performed appropriately and rigorously? 

Reviewer #1: (No Response)

Reviewer #2: Yes

4. Have the authors made all data underlying the findings in their manuscript fully available?

Reviewer #1: (No Response)

Reviewer #2: Yes

5. Is the manuscript presented in an intelligible fashion and written in standard English?

Reviewer #1: (No Response)

Reviewer #2: Yes

6. Review Comments to the Author

Reviewer #1: I have read the responses to my earlier comments and the revised manuscript and am satisfied. As I said earlier, I truly appreciate the authors' transparency and commitment to high-quality and pre-registered research. I will now add that I also appreciate their willingness to take feedback so constructively.

Reviewer #2: Thanks to the authors for their responses and thorough revisions. I have just a few more comments, mostly minor.

Major comments:

In "2.3 Willingness to pay: Random price auctions" starting on line 253, how were households identified? This is important, as, if not random, using Auction 1 on the first 69 households and Auction 2 on the last 93 households can produce biased results.

Line 206 “Study Setting": How does this setting impact the interpretation of results? What population are we learning about?

Paragraph starting on line 373: Even though you use a status-quo scenario and compare alternatives to this, with 124 possible scenarios but only 162 respondents, are you powered enough to find anything? And if so, is the N so small it may be biased?

Minor comments:

Line 61: “Access to piped water has expanded significantly (Murray et al. 2020), but, as of 2020, just over 83% of urban households and 42% of rural households received piped water services (WHO/UNICEF 2021)”

Is this referring to the 2 billion households using contaminated water or all total households?

Line 101-112 onward should be present tense. Same with line 137.

Line 600: “though, notably, the coefficient on a price of ₹9 is large and negative.” What does this mean? What is the interpretation?

Line 651 : “Our findings led to a failure to reject alternative hypothesis 2, suggesting that the effect of missing out on small discounts had modest but larger negative effects on demand.” How does this suggestion/interpretation depend on possible biases in estimates?

7. PLOS authors have the option to publish the peer review history of their article (what does this mean?). If published, this will include your full peer review and any attached files.

Reviewer #1: **Yes: **Joseph Cook

Reviewer #2: No

---

## [Author Response · Author response to Decision Letter 1]

6 Mar 2023

We have responded to all remaining reviewer comments in the attached document "Response to Reviewers.docx". Those responses are reproduced below (our responses follow a "-"): 

Reviewer #1: I have read the responses to my earlier comments and the revised manuscript and am satisfied. As I said earlier, I truly appreciate the authors' transparency and commitment to high-quality and pre-registered research. I will now add that I also appreciate their willingness to take feedback so constructively.

- We appreciated your constructive comments! Thank you for your careful reading of our manuscript.

Reviewer #2: Thanks to the authors for their responses and thorough revisions. I have just a few more comments, mostly minor.

Major comments:

In "2.3 Willingness to pay: Random price auctions" starting on line 253, how were households identified? This is important, as, if not random, using Auction 1 on the first 69 households and Auction 2 on the last 93 households can produce biased results.

- Thank you for this comment. We have provided added clarity on pages 14-5 lines 301-11: 

- “Households were selected for the study during a listing process in which members of the study enumeration team walked through eligible neighborhoods and screened potential households for inclusion. Eligible households were not selected into Auction 1 versus Auction 2 by randomization, but rather by listing order, thus it is possible that household characteristics differ between the two groups, which could potentially introduce bias of uncertain sign in our comparisons of the auctions. Table 3 presents formal tests of differences in baseline characteristics between the two auction groups, and finds significant differences, indicating the importance of testing for sensitivity of results to controlling for these baseline characteristics so as to quantify potential bias at least from these observed differences. In Table 5 we present the main auction comparisons without versus with these baseline controls and find that adding these controls has only minor effects on the estimated coefficients and does not change our overall conclusions.”

Line 206 “Study Setting": How does this setting impact the interpretation of results? What population are we learning about?

- We agree with the need for additional clarity here about the population under study. We have revised pages 9-10 lines 226-8 to read: 

- “Because this sample is slightly poorer than the average Bihari household, our results may have a slight downward bias with regard to product adoption, though they would likewise not be generalizable to the poorest of the poor households in the region.”

Paragraph starting on line 373: Even though you use a status-quo scenario and compare alternatives to this, with 124 possible scenarios but only 162 respondents, are you powered enough to find anything? And if so, is the N so small it may be biased?

- Power is a potential concern for our study. Thus, we have updated the last sentence of the paragraph you highlighted to include the following on page 19 lines 405-13: 

- “Following Johnson et al. (2013), we eliminated all 'implausible' alternative product characteristic alternative scenarios that were strongly dominated by the status quo choice (e.g., “a delivered water product for a positive price that tastes bad, requires request for delivery, can make respondents sick, and is not used by neighbors”) leaving a total of 124 possible alternatives to the status quo to be presented to 162 respondents (3 times each, totaling 486 choices) over two rounds. Although this is a large number of alternatives relative to the sample size, the power implications of this are mitigated by following the standard methods in this literature of estimating only a small number of choice set parameters (eight). Our resulting confidence intervals confirm that we had sufficient power to estimate meaningful effect sizes.”

Minor comments:

Line 61: “Access to piped water has expanded significantly (Murray et al. 2020), but, as of 2020, just over 83% of urban households and 42% of rural households received piped water services (WHO/UNICEF 2021)”

Is this referring to the 2 billion households using contaminated water or all total households?

- Thank you for pointing out the vague language. We have revised the passage on page 3 line 63 to read: 

- “Access to piped water has expanded significantly (Murray et al. 2020), but, as of 2020, just over 83% of urban households worldwide and 42% of rural households received piped water services (WHO/UNICEF 2021).”

Line 101-112 onward should be present tense. Same with line 137.

- We have amended the relevant paragraphs, now pages 4-5 lines 101-110 and page 6 line 140, to read: 

- “This study has multiple aims designed to inform future WASH studies and programs on demand and preferences for clean water. The first is to examine willingness to pay (WTP) for a treated water delivery service among residents to assess product pricing both in real terms and as a share of household income to inform the literature on product pricing. Second, we seek to examine the possible role of subsidies in increasing short-term demand and product uptake to inform a larger, randomized controlled trial that examines the potential for subsidies to lead to price anchoring and/or positive learning. Third, we aim to identify specific product characteristics that are subject to experiential learning among new customers to add to existing knowledge on how preferences for water products change over time. In regions where public utilities are unlikely (for the present) to enter “last-mile” communities, a more complete understanding of the acceptability and affordability of treated and delivered water is essential for policymaking on safe drinking water solutions.”

- And 

- “Among these studies, some of the most important water product characteristics are price, taste, smell, health benefits (both real and perceived), convenience of use (including time required), durability and aesthetics.”

- Respectively. 

Line 600: “though, notably, the coefficient on a price of ₹9 is large and negative.” What does this mean? What is the interpretation?

- Since this result was insignificant, we have struck the line from the manuscript (see line 626).

Line 651 : “Our findings led to a failure to reject alternative hypothesis 2, suggesting that the effect of missing out on small discounts had modest but larger negative effects on demand.” How does this suggestion/interpretation depend on possible biases in estimates?

- This is an important point, and one for which we include the following revised caveat on page 40 lines 680-83: 

- “Our findings led to a failure to reject alternative hypothesis 2, suggesting that the effect of missing out on small discounts had modest but larger negative effects on demand. We caution that there may be unobservable sources of bias inherent to those who were sampled for Auction 2 based on our non-random household listing procedure. However, the direction of potential bias is unclear.”

---

## [Editor Report · Decision Letter 2]

21 Mar 2023

Product preferences and willingness to pay for potable water delivery: Experimental evidence from rural Bihar, India

PONE-D-22-05048R2

Dear Dr. Cameron,

We’re pleased to inform you that your manuscript has been judged scientifically suitable for publication and will be formally accepted for publication once it meets all outstanding technical requirements.

Kind regards,

Ilke Onur, Ph.D.

Academic Editor

PLOS ONE

---

## [Editor Report · Acceptance letter]

28 Mar 2023

PONE-D-22-05048R2 

Product preferences and willingness to pay for potable water delivery: Experimental evidence from rural Bihar, India 

Dear Dr. Cameron:

I'm pleased to inform you that your manuscript has been deemed suitable for publication in PLOS ONE. Congratulations! Your manuscript is now with our production department. 

Kind regards, 

on behalf of

Dr. Ilke Onur 

Academic Editor

PLOS ONE